# Response-based outcome predictions and confidence regulate feedback processing and learning

Romy Frömer[1,2]*, Matthew R Nassar[2], Rasmus Bruckner[3,4,5], Birgit Stürmer[6], Werner Sommer[1], Nick Yeung[7]

[1]Humboldt-Universität zu Berlin, Berlin, Germany; [2]Brown University, Providence, United States; [3]Freie Universität Berlin, Berlin, Germany; [4]Max Planck School of Cognition, Leipzig, Germany; [5]International Max Planck Research School LIFE, Berlin, Germany; [6]International Psychoanalytic University, Berlin, Germany; [7]University of Oxford, Oxford, United Kingdom

**Abstract** Influential theories emphasize the importance of predictions in learning: we learn from feedback to the extent that it is surprising, and thus conveys new information. Here, we explore the hypothesis that surprise depends not only on comparing current events to past experience, but also on online evaluation of performance via internal monitoring. Specifically, we propose that people leverage insights from response-based performance monitoring – outcome predictions and confidence – to control learning from feedback. In line with predictions from a Bayesian inference model, we find that people who are better at calibrating their confidence to the precision of their outcome predictions learn more quickly. Further in line with our proposal, EEG signatures of feedback processing are sensitive to the accuracy of, and confidence in, post-response outcome predictions. Taken together, our results suggest that online predictions and confidence serve to calibrate neural error signals to improve the efficiency of learning.

*For correspondence: romy_fromer@brown.edu

**Competing interests:** The authors declare that no competing interests exist.

## Introduction

Feedback is crucial to learning and adaptation. Across domains it is thought that feedback drives learning to the degree that it is unexpected and, hence, provides new information, for example in the form of prediction errors that express the discrepancy between actual and expected outcomes (*McGuire et al., 2014*; *Yu and Dayan, 2005*; *Behrens et al., 2007*; *Diederen and Schultz, 2015*; *Diederen et al., 2016*; *Pearce and Hall, 1980*; *Faisal et al., 2008*; *Sutton and Barto, 1998*; *Wolpert et al., 2011*). Yet, the same feedback can be caused by multiple sources: we may be wrong about what is the correct thing to do, or we may know what to do but accidentally still do the wrong thing (*McDougle et al., 2016*). When we know we did the latter, we should discount learning about the former (*McDougle et al., 2019*; *Parvin et al., 2018*). Imagine for instance learning to throw darts. You know the goal you want to achieve – hit the bullseye – and you might envision yourself performing the perfect throw to do so. However, you find that the throw you performed as intended missed the target entirely and did not yield the desired outcome: In this case, you should adjust what you believe to be the right angle to hit the bullseye, based on *how* you missed that last throw. On a different throw you might release the dart at a different angle than intended and thus anticipate the ensuing miss: In this case, you may not want to update your beliefs on what is the right angle of throw. How do people assign credit to either of these potential causes of feedback when learning how to perform a new task? How do they regulate how much to learn from a given feedback depending on how much they know about its causes?

Performance monitoring, that is the internal evaluation of one's own actions, could reduce surprise about feedback and uncertainty about its causes by providing information about execution errors. For instance in the second dart throw example, missing the target may be unsurprising if performance monitoring detected that, for example, the dart was released differently than desired (*Figure 1A*). In simple categorical choices, people are often robustly aware of their response errors (*Maier et al., 2011*; *Yeung et al., 2004*; *Riesel et al., 2013*; *Maier et al., 2012*) and this awareness is reflected in neural markers of error detection (*Murphy et al., 2015*). Although errors are often studied in simple categorization tasks in which responses are either correct or incorrect, in many tasks, errors occur on a graded scale (e.g. a dart can miss the target narrowly or by a large margin), and both error detection, as well as feedback processing are sensitive to error magnitude (*Luft et al., 2014*; *Ulrich and Hewig, 2014*; *Frömer et al., 2016a*; *Arbel and Donchin, 2011*). People are even able to report gradual errors reasonably accurately (*Kononowicz et al., 2019*; *Akdoğan and Balcı, 2017*; *Kononowicz and van Wassenhove, 2019*).

This ability may be afforded by reliance on internal models to predict the outcome of movements (*Wolpert and Flanagan, 2001*), for example, based on an efference copy of a motor command. These predictions could help discount execution errors in learning from feedback. In fact, if these predictions perfectly matched the execution error that occurred, the remaining mismatch between predicted and obtained feedback (sensory prediction error) could serve as a reliable basis for adaptation and render feedback maximally informative about the mapping from actions to outcomes (*Figure 1B*).

Although participants are able to evaluate their own performance reasonably well, error detection is far less certain than outlined in the ideal scenario above, and the true cause of feedback often remains uncertain to some extent. People are critically sensitive to uncertainty, and learn more from feedback when they expect it to be more informative (*McGuire et al., 2014*; *Schiffer et al., 2017*; *Bland and Schaefer, 2012*; *Nassar et al., 2010*; *O'Reilly, 2013*). Uncertainty about what caused a given feedback inevitably renders it less informative, similar to decreases in reliability, and this uncertainty should be taken into account when learning from it. Confidence could support such adaptive learning from feedback by providing a read-out of the subjective precision of predicted outcomes (*Nassar et al., 2010*; *Vaghi et al., 2017*; *Meyniel et al., 2015*; *Pouget et al., 2016*), possibly relying on shared neural correlates of confidence with error detection (*Boldt and Yeung, 2015*; *van den Berg et al., 2016*). Similar to its role in regulating learning of transition probabilities (*Meyniel et al., 2015*; *Meyniel and Dehaene, 2017*), information seeking/exploration in decision making (*Desender et al., 2018a*; *Boldt et al., 2019*), and hierarchical reasoning (*Sarafyazd and Jazayeri, 2019*), people could leverage confidence to calibrate their use of online predictions. In line with this suggestion, people learn more about advice givers when they are more confident in the choices that advice is about (*Carlebach and Yeung, 2020*). In the throwing example above, the more confident you are about the exact landing position of the dart, the more surprised you should be when you find that landing position to be different: The more confident you are, the more evidence you have that your internal model linking angles to landing positions is wrong, and the more information you get about how this model is wrong. Thus, you should learn more when you are more confident. However, this reasoning assumes that your predictions are in fact more precise when you are more confident, i.e., that your confidence is well calibrated (*Figure 1B*).

In the present study, we tested the hypothesis that performance monitoring – error detection and confidence (*Yeung and Summerfield, 2012*) – adaptively regulates learning from feedback. This hypothesis predicts that error detection and confidence afford better learning, with confidence mediating the relationship between outcome predictions and feedback, and that learning is compromised when confidence is mis-calibrated (*Figure 1C*). It further predicts that established neural correlates of feedback processing, such as the feedback-related negativity (FRN) and the P3a (*Ullsperger et al., 2014a*), should integrate information about post-response outcome predictions and confidence. That is to say, an error that could be predicted based on internal knowledge of how an action was executed should *not* yield a large surprise (P3a) or reward prediction error (FRN) signal in response to an external indicator of the error (feedback). However, any prediction error should be *more* surprising when predictions were made with higher confidence. We formalize our predictions using a Bayesian model of learning and test them using behavioral and EEG data in a modified time-estimation task.

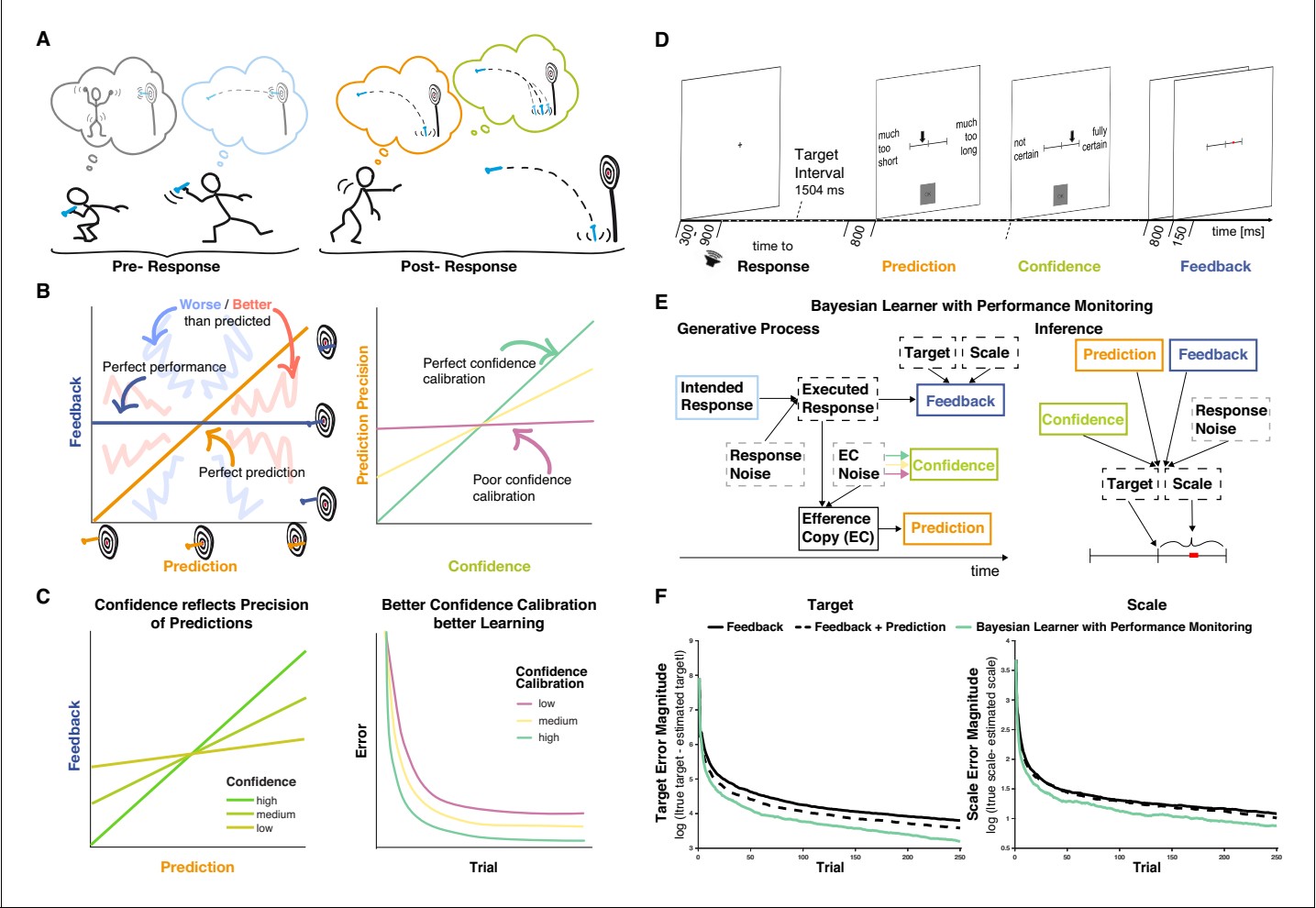

**Figure 1.** Interactions between performance monitoring and feedback processing. (**A**) Illustration of dynamic updating of predicted outcomes based on response information. Pre-response the agent aims to hit the bullseye and selects the action he believes achieves this goal. Post-response the agent realizes that he made a mistake and predicts to miss the target entirely, being reasonably confident in his prediction. In line with his prediction and thus unsurprisingly the darts hits the floor. (**B**) Illustration of key concepts. Left: The feedback received is plotted against the prediction. Performance and prediction can vary in their accuracy independently. Perfect performance (zero deviation from the target, dark blue line) can occur for accurate or inaccurate predictions and any performance, including errors, can be predicted perfectly (predicted error is identical to performance, orange line). When predictions and feedback diverge, outcomes (feedback) can be better (closer to the target, area highlighted with coarse light red shading) or worse (farther from the target, area highlighted with coarse light blue shading) than predicted. The more they diverge the less precise the predictions are. Right: The precision of the prediction is plotted against confidence in that prediction. If confidence closely tracks the precision of the predictions, that is if agents know when their predictions are probably right and when they're not, confidence calibration is high (green). If confidence is independent of the precision of the predictions, then confidence calibration is low. (**C**) Illustration of theoretical hypotheses. Left: We expect the correspondence between predictions and Feedback to be stronger when confidence is high and to be weaker when confidence is low. Right: We expect that agents with better confidence calibration learn better. (**D**) Trial schema. Participants learned to produce a time interval by pressing a button following a tone with their left index finger. Following each response, they indicated on a visual analog scale in sequence the estimate of their accuracy (anchors: 'much too short' = 'viel zu kurz' to 'much too long' = 'viel zu lang') and their confidence in that estimate (anchors: 'not certain' = 'nicht sicher' to 'fully certain' = 'völlig sicher') by moving an arrow slider. Finally, feedback was provided on a visual analog scale for 150 ms. The current error was displayed as a red square on the feedback scale relative to the target interval indicated by a tick mark at the center (Target, t) with undershoots shown to the left of the center and overshoots to the right, and scaled relative to the feedback anchors of -/+1 s (Scale, s; cf. **E**). Participants are told neither Target nor Scale and instead need to learn them based on the feedback. (**E**) Bayesian Learner with Performance Monitoring. The learner selects an intended response (i) based on the current estimate of the Target. The Intended Response and independent Response Noise produce the Executed Response (r). The Efference Copy (c) of this response varies in its precision as a function of Efference Copy Noise. It is used to generate a Prediction as the deviation from the estimate of Target scaled by the estimate of Scale. The Efference Copy Noise is estimated and expressed as Confidence (co), approximating the precision of the Prediction. Learners vary in their Confidence Calibration (cc), that is, the precision of their predictions, and higher Confidence Calibration (arrows: green >yellow > magenta) leads to more reliable translation from Efference Copy precision to Confidence. Feedback is provided according to the Executed Response and depends on the Target and Scale, which are unknown to the learner. Target and Scale are inferred

*Figure 1 continued on next page*

*Figure 1 continued*

based on Feedback (f), Response Noise, Prediction, and Confidence. Variables that are observable to the learner are displayed in solid boxes, whereas variables that are only partially observable are displayed in dashed boxes. (F) Target and scale error (absolute deviation of the current estimates from the true values) for the Bayesian learner with Performance monitoring (green, optimal calibration), a Feedback-only Bayesian Learner (solid black), and a Bayesian Learner with Outcome Prediction (dashed black).

## Results

### Rationale and approach

Our hypothesis that performance monitoring regulates adaptive learning from feedback makes two key behavioral predictions (*Figure 1C*): (1) The precision of outcome predictions (i.e. the correlation between predicted and actual outcomes) should increase with confidence. (2) Learners with superior calibration of confidence to the precision of their outcome predictions should learn more quickly. Our hypothesis further predicts that feedback processing will be critically modulated by an agent's outcome prediction and confidence. We tested these predictions mechanistically using computational modeling and empirically based on behavioral and EEG data from 40 participants performing a modified time-estimation task (*Figure 1D*). In comparison to darts throwing as used in our example, the time estimation task requires a simple response – a button press – such that errors map onto a single axis that defines whether the response was provided too early, timely, or too late and by how much. These errors can be mapped onto a feedback scale and, just as in the darts example where one learns the correct angle and acceleration to hit the bullseye, participants here can learn the target timing interval. In addition to requiring participants to learn and produce a precisely timed action on each trial, our task also included two key measurements that allowed us to better understand how performance monitoring affects feedback processing: (1) Participants were required to predict the feedback they would receive on each trial and indicate it on a scale visually identical to the feedback scale (*Figure 1D*, Prediction) and (2) Participants indicated their degree of confidence in this prediction (*Figure 1D*, Confidence). Only following these judgments would they receive feedback about their time estimation performance.

### A mechanism for performance monitoring-augmented learning

As a demonstration of proof of the hypothesized learning principles, we implemented a computational model that uses performance monitoring to optimize learning from feedback in that same task (*Figure 1E*). The agent's goal is to learn the mapping between its actions and their outcomes (sensory consequences) in the time-estimation task, wherein feedback on an initially unknown scale must be used to learn accurately timed actions. Learning in this task is challenged in two ways: First, errors signaled by feedback include contributions of response noise, for example, through variability in the motor system or in the representations of time (*Kononowicz and van Wassenhove, 2019*; *Balci et al., 2011*). Second, the efference copy of the executed response (or the estimate of what was done) varies in its precision. To overcome these challenges, the agent leverages performance monitoring: It infers the contribution of response noise to a given outcome based on an outcome prediction derived from the efference copy, and the degree of confidence in its prediction based on an estimate of the current efference copy noise. The agent then weighs Prediction and Intended Response as a function of Confidence and Response Noise when updating beliefs about the Target and the Scale based on Feedback.

We compare this model to one that has no insights into its trial-by-trial performance, but updates based on feedback and its fidelity due to response noise alone (Feedback), and another model that has insights into its trial-by-trial performance allowing it to generate predictions, and into the average precision of its predictions, but not the precision of its current prediction (Feedback + Prediction). We find that performance improves as the amount of insight into the agent's performance increases (*Figure 1F*): The optimally calibrated Bayesian learner with performance monitoring outperforms both other models. Further, in line with our behavioral predictions, we find in this model that confidence varies with the precision of predictions (*Figure 2A*, *Figure 2—figure supplement 1*) and, when varying the fidelity of confidence as a read-out of precision (Confidence Calibration), agents with superior Confidence Calibration learn better (*Figure 2B*, *Figure 2—*

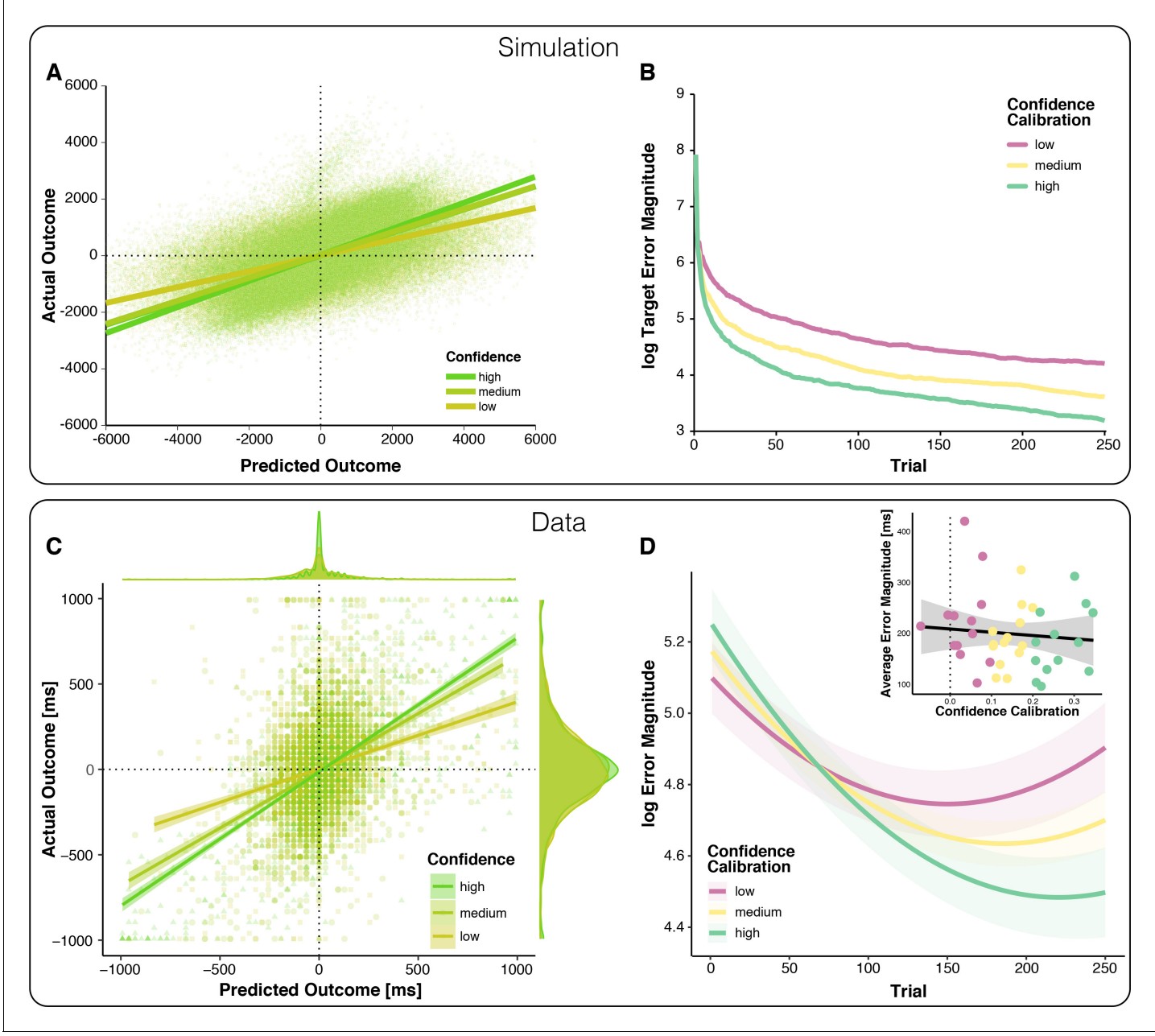

**Figure 2.** Relationships between outcome predictions and actual outcomes in the model and observed data (top vs.bottom). (**A**) Model prediction for the relationship between Prediction and actual outcome (Feedback) as a function of Confidence. The relationship between predicted and actual outcomes is stronger for higher confidence. Note that systematic errors in the model's initial estimates of target (overestimated) and scale (underestimated) give rise to systematically late responses, as well as underestimation of predicted outcomes in early trials, visible as a plume of datapoints extending above the main cloud of simulated data. (**B**) The model-predicted effect of Confidence Calibration on learning. Better Confidence Calibration leads to better learning. (**C**) Observed relationship between predicted and actual outcomes. Each data point corresponds to one trial of one participant; all trials of all participants are plotted together. Regression lines are local linear models visualizing the relationship between predicted and actual error separately for high, medium, and low confidence. At the edges of the plot, the marginal distributions of actual and predicted errors are depicted by confidence levels. (**D**) Change in error magnitude across trials as a function of confidence calibration. Lines represent LMM-predicted error magnitude for low, medium and high confidence calibrations, respectively. Shaded error bars represent corresponding SEMs. Note that the combination of linear and quadratic effects approximates the shape of the learning curves, better than a linear effect alone, but predicts an exaggerated uptick in errors toward the end, ***Figure 2—figure supplement 3***. Inset: Average Error Magnitude for every participant plotted as a function of Confidence Calibration level. The vast majority of participants show positive confidence calibration. The regression line represents a local linear model fit and the error bar represents the standard error of the mean.

The online version of this article includes the following figure supplement(s) for figure 2:

*Figure 2 continued on next page*

*Figure 2 continued*

**Figure supplement 1.** Model comparison.
**Figure supplement 2.** Predictions and Confidence improve as learning progresses.
**Figure supplement 3.** Running average log error magnitude across trials.

*figure supplement 1*). We next sought to test whether participants' behavior likewise displays these hallmarks of our hypothesis.

## Confidence reflects precision of outcome predictions

To test the predictions of our model empirically, we examined behavior of 40 human participants performing the modified time-estimation task. To test whether the precision of outcome predictions increases with confidence, we regressed participants' *signed* timing production errors (signed error magnitude; scale: undershoot [negative] to overshoot [positive]) on their *signed* outcome predictions (Predicted Outcome; same scale as for signed error magnitude), Confidence, Block, as well as their interactions. Our results support our first behavioral prediction (*Table 1*): As expected, predicted outcomes and actual outcomes were positively correlated, indicating that participants could broadly indicate the direction and magnitude of their errors. Crucially, this relationship between predicted and actual outcomes was stronger for predictions made with higher confidence (*Figure 2C*).

In addition to this expected pattern, we found that both outcome predictions, as well as confidence calibration, improved across blocks, suggestive of learning at the level of performance monitoring (*Figure 2—figure supplement 2*). Note however that participants tended to bias their predictions toward the center of the scale in early blocks, when they had little knowledge about the target interval and could thus determine neither over- vs. undershoots nor their magnitude. This strategic behavior may give rise to the apparent improvements in performance monitoring.

To test more directly our assumption that Confidence tracks the precision of predictions, we followed up on these findings with a complementary analysis of Confidence as the dependent variable and tested how it relates to the precision of predictions (absolute discrepancy between predicted and actual outcome, see sensory prediction error, SPE below), the precision of performance (error magnitude), and how those change across blocks (*Table 2*). Consistent with our assumption that Confidence tracks the precision of predictions, we find that it increases as the discrepancy between

**Table 1.** Relations between actual performance outcome (signed error magnitude), predicted outcome, confidence in predictions and their modulations due to learning across blocks of trials.

| | Signed error magnitude | | | | |
|---|---|---|---|---|---|
| *Predictors* | *Estimates* | *SE* | *CI* | *t* | *p* |
| Intercept | 4.63 | 9.99 | −14.94–24.20 | 0..46 | 6.427e-01 |
| Predicted Outcome | 523.99 | 29.66 | 465..86–582.12 | 17.67 | **7.438e-70** |
| Block | 29.47 | 8.12 | 13..56–45.37 | 3..63 | **2.832e-04** |
| Confidence | −27.07 | 11.05 | −48.73 − −5.42 | −2..45 | **1.428e-02** |
| Predicted Outcome: Block | −149.70 | 21.90 | −192.62 − −106.78 | −6..84 | **8.145e-12** |
| Predicted Outcome: Confidence | 322.56 | 27.31 | 269.03–376.09 | 11.81 | **3.477e-32** |
| Block: Confidence | −25.52 | 9..15 | −43.46 − −7.58 | −2..79 | **5.297e-03** |
| Predicted Outcome: Block: Confidence | 90.68 | 33.65 | 24.73–156.64 | 2..69 | **7.043e-03** |
| **Random effects** | | | **Model Parameters** | | |
| Residuals | 54478.69 | | N | 40 | |
| Intercept | 3539.21 | | Observations | 9996 | |
| Confidence | 2813.79 | | log-Likelihood | −68816.092 | |
| Predicted Outcome | 22357.33 | | Deviance | 137632.185 | |

Formula: Signed error magnitude ~Predicted Outcome*Block*Confidence+(Confidence +Predicted Outcome+Block|participant); Note: ':'' indicates interactions between predictors.

**Table 2.** Relations of confidence with the precision of prediction and the precision of performance and changes across blocks.

| | Confidence | | | | |
|---|---|---|---|---|---|
| Predictors | Estimates | SE | CI | t | p |
| (Intercept) | 0.26 | 0.04 | 0.18–0.33 | 6.35 | 2.187e-10 |
| Block | 0.05 | 0.02 | 0.02–0.08 | 3.05 | 2.257e-03 |
| Sensory Prediction Error (SPE) | −0.44 | 0.04 | −0.52 − −0.36 | −10.84 | 2.289e-27 |
| Error Magnitude (EM) | 0.17 | 0.05 | 0.08–0.27 | 3.73 | 1.910e-04 |
| Block: SPE | −0.08 | 0.04 | −0.15 − −0.00 | −1.99 | 4.642e-02 |
| Block: EM | 0.15 | 0.05 | 0.05–0.25 | 3.07 | 2.167e-03 |
| Random effects | | | Model Parameters | | |
| Residuals | 0.12 | | N | 40 | |
| Intercept | 0.06 | | Observations | 9996 | |
| SPE | 0.03 | | log-Likelihood | −3640.142 | |
| Error Magnitude | 0.06 | | Deviance | 7280.284 | |
| Block | 0.01 | | | | |
| Error Magnitude: Block | 0.04 | | | | |

Formula: Confidence ~ (SPE +Error Magnitude)*Block+(SPE +Error Magnitude *Block|participant); Note: ':" indicates interactions between predictors.

predicted and actual outcome decreases. Confidence was also higher for larger errors, presumably because their direction (i.e. overshoot or undershoot) is easier to judge. The relationships with both the precision of the prediction and error magnitude changed across blocks, and confidence increased across blocks as well.

To test whether these effects reflect monotonic increases in confidence and its relationships with prediction error and error magnitude, as expected with learning, we fit a model with block as a categorical predictor and SPE and Error Magnitude nested within blocks (*Supplementary file 1*). We found that confidence increased numerically from each block to the next, with significant differences between block 1 and 2, as well as block 3 and 4. Its relationship to error magnitude was reduced in the first block compared to the remaining blocks and enhanced in the final two blocks compared to the remaining blocks. These findings are thus consistent with learning effects. While the precision of predictions was more strongly related to confidence in the final block compared to the remaining blocks, it was not less robustly related in the first block, and instead somewhat weaker in the third block. This pattern is thus not consistent with learning. Importantly, whereas error magnitude was robustly related to confidence only in the last two blocks, the precision of the prediction was robustly related to confidence throughout.

Having demonstrated that, across individuals, confidence reflects the precision of their predictions (via the correlation with SPE), we next quantified this relationship for each participant separately as an index of their confidence calibration. While quantifying the relationship, we controlled for changes in performance across blocks, and to ease interpretation, we sign-reversed the obtained correlations so that higher values correspond to better confidence calibration. We next tested our hypothesis that confidence calibration relates to learning.

## Superior calibration of confidence judgments relates to superior learning

To empirically test our second behavioral prediction, that people with better confidence calibration learn faster, we modeled log-transformed trial-wise error magnitude as a function of Trial (linear and quadratic effects to account for non-linearity in learning, that is stronger improvements in the beginning), Confidence Calibration for each participant (*Figure 2D* inset), and their interaction (*Table 3*). As expected, Confidence Calibration interacted significantly with the linear Trial component, that is with learning (*Figure 2D*). Thus, participants with better confidence calibration showed greater performance improvements during the experiment. Importantly, Confidence Calibration did not significantly correlate with overall performance (*Figure 2D* inset), supporting the assumption that

**Table 3.** Confidence calibration modulation of learning effects on performance.

| | log Error Magnitude | | | | |
|---|---|---|---|---|---|
| *Predictors* | *Estimates* | *SE* | *CI* | t | p |
| (Intercept) | 5.17 | 0.06 | 5.05–5.30 | 80.74 | **0.000e + 00** |
| Confidence Calibration | 0.58 | 0.58 | −0.57–1.72 | 0.99 | 3.228e-01 |
| Trial (linear) | −0.59 | 0.07 | −0.72 − −0.45 | −8..82 | **1.197e-18** |
| Trial (quadratic) | 0.16 | 0.02 | 0.11–0.20 | 6.80 | **1.018e-11** |
| Trial (linear): Confidence Calibration | −0.86 | 0.32 | −1.48 − −0.24 | −2.72 | **6.467e-03** |
| **Random effects** | | | **Model Parameters** | | |
| Residuals | 1.18 | | N | 40 | |
| Intercept | 0..12 | | Observations | 9996 | |
| Trial (linear) | 0..03 | | log-Likelihood | −15106.705 | |
| | | | Deviance | 30213.411 | |

Formula: log Error Magnitude ~ (Confidence Calibration* Trial(linear)+Trial(quadratic) + (Trial(linear)|participant)); Note: ':' indicates interactions between predictors.

confidence calibration relates to learning (performance change), rather than performance per se. Confidence calibration was also not correlated with individual differences in response variance (r = -2.07e-4, 95% CI = [−0.31, 0.31], p=0.999), and the interaction of confidence calibration and block was robust to controlling for running average response variance (*Supplementary file 2*).

Thus, taken together, our model simulations and behavioral results align with the behavioral predictions of our hypothesis: Participants' outcome predictions were better related to actual outcomes when those outcome predictions were made with higher confidence, and individuals with superior confidence calibration showed better learning.

## Outcome predictions and confidence modulate feedback signals and processing

At the core of our hypothesis and model lies the change in feedback processing as a function of outcome predictions and confidence. It is typically assumed that learning relies on prediction errors, and signatures of prediction errors have been found in scalp-recorded EEG signals. Before testing directly how feedback is processed, as reflected in distinct feedback related ERP components, we will show how these prediction errors vary over time, and as a function of confidence.

We dissociate three signals that can be processed to evaluate feedback (*Figure 3A*): The objective magnitude of the error (Error Magnitude) reflects the degree to which performance needs to be adjusted regardless of whether that error was predicted or not. The reward prediction error (RPE), thought to drive reinforcement learning, indexes whether the outcome of a particular response was better or worse than expected. The sensory prediction error (SPE), thought to underlie forward model-based and direct policy learning in the motor domain (*Hadjiosif et al., 2020*), indexes whether the outcome of a particular response was close to or far off the predicted one. To illustrate the difference between the two prediction errors, one might expect to miss a target 20 cm to the left but find the arrow misses it 20 cm to the right instead. There is no RPE, as the actual outcome is exactly as good or bad as the predicted one, however, there is a large SPE, because the actual outcome is very different from the predicted one.

Our hypothesis holds that predictions should help discount noise in the error signal and more so for higher confidence. Prediction errors should thus be smaller than error magnitude and particularly so when confidence is higher. We find that this is true in our data (*Figure 3B*, *Supplementary file 3* and *4*, note that unlike SPE, by definition RPE cannot be larger than error magnitude and that its magnitude, but not sign varies robustly with confidence).

To examine changes in these error signals with trial-to-trial changes in confidence and learning, we regressed each of these signals onto Confidence, Block, and their interaction (*Supplementary file 5*, *Figure 3B*). Consistent with our assumption that confidence tracks the precision of predictions, SPE decreased as confidence increased (b = −71.20, p>0.001), but there were

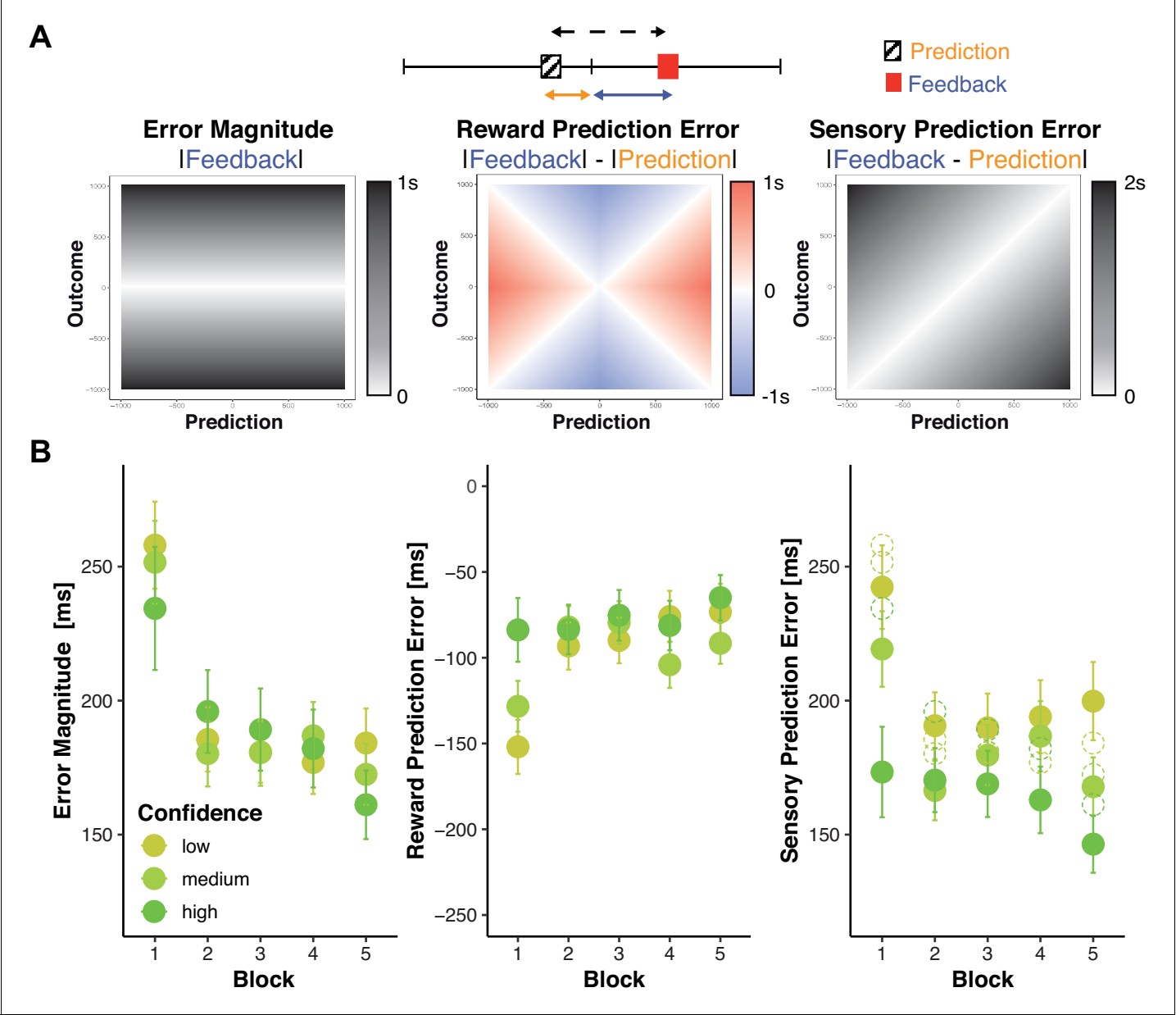

**Figure 3.** Changes in objective and subjective feedback. (A) Dissociable information provided by feedback. An example for a prediction (hatched box) and a subsequent feedback (red box) are shown overlaid on a rating/feedback scale. We derived three error signals that make dissociable predictions across combinations of predicted and actual outcomes. The solid blue line indicates Error Magnitude (distance from outcome to goal). As smaller errors reflect greater rewards, we computed Reward Prediction Error (RPE) as the signed difference between negative Error Magnitude and the negative predicted error magnitude (solid orange line, distance from prediction to goal). Sensory Prediction Error (SPE, dashed line) was quantified as the absolute discrepancy between feedback and prediction. Values of Error Magnitude (left), RPE (middle), and SPE (right) are plotted for all combinations of prediction (x-axis) and outcome (y-axis) location. (B) Predictions and confidence associate with reduced error signals. Average error magnitude (left), Reward Prediction Error (center), and Sensory Prediction Error (right) are shown for each block and confidence tercile. Average prediction errors are smaller than average error magnitudes (dashed circles), particularly for higher confidence.

no significant main effects on error magnitude or reward prediction error. However, Confidence significantly interacted with Block on all variables (Error Magnitude: b = 30.09, p<0.001, RPE: b = −64.48, p<0.001, SPE: b = 16.99, p=0.005), such that in the first block, increased Confidence is associated with smaller Error Magnitudes, less negative RPE, as well as smaller SPE. All error signals further decreased significantly across blocks (Error Magnitude: b = - 37.10, p<0.001, RPE: b = 36.26,

p<0.001, SPE: b = - 17.54, p<0.001, block wise comparisons significant only from block 1 to 2 *Supplementary file 6*). These parallel patterns might emerge because prediction errors are derived from and thus might covary with error magnitude. To test whether changes in prediction errors were primarily driven by improvements in error magnitude rather than predictions, we reran the previous RPE and SPE models with error magnitude as a covariate (*Supplementary file 7*). Controlling for error magnitude notably reduced linear block effects on RPE (b = 36.26 to b = 10.3). It further eliminated block effects on SPE: b = −17.54 to b = 3.65, p=0.274, as well as the interaction of confidence and Block (b = 0.10, p=0.984), while the hypothesized main effect of Confidence prevailed (b = −60.12, p<0.001).

In summary, we find that all error signals decrease across blocks as performance improves. Although higher confidence is associated with smaller error signals in all three variables early in learning, across all blocks we find that confidence only has a consistent relationship with smaller sensory prediction errors.

Taken together, these results are consistent with our hypothesis that outcome predictions and confidence optimize feedback processing. Accordingly, we predicted that participants' internal evaluations would modulate feedback processing as indexed by distinct feedback-related potentials in the EEG: the feedback-related negativity (FRN), P3a and P3b. Thus, the amplitude of a canonical index of signed RPE (*Holroyd and Coles, 2002*), the FRN, should increase to the extent that outcomes were worse than predicted, that is, with more negative-going RPE. P3a amplitude, a neural signature of surprise (*Polich, 2007*), should increase with the absolute difference between participants' outcome predictions and actual outcomes (i.e. with SPE) and be enhanced in trials in which participants indicated higher confidence in their outcome predictions. To further explore the possible role of performance monitoring in learning, we also tested the joint effects of our experimental variables on the P3b as a likely index of learning (*Fischer and Ullsperger, 2013*).

If participants did not take their predictions into account, ERP amplitudes should scale with the actual error magnitude reflected in the feedback (Error Magnitude). Note that both RPE and SPE are equivalent to Error Magnitude in the special case where predicted errors are zero (*Figure 3A*), and thus Error Magnitude can be thought of as the default RPE and SPE that would arise if an individual predicted perfect execution on each trial. Thus, if participants did not take knowledge of their own execution errors into account, their FRN and P3a should both simply reflect Error Magnitude. A key advantage of our experimental design is that RPE, SPE, and Error Magnitude vary differentially as a function of actual and predicted outcomes (*Figure 3A*), which allowed us to test our predictions by examining whether ERP amplitudes are modulated by prediction errors (SPE and RPE) and Confidence, while controlling for other factors including Error Magnitude.

## Reward prediction error modulates the feedback-related negativity

The feedback-related negativity (FRN) is an error-sensitive ERP component with a fronto-central scalp distribution that peaks between 230 and 330 ms following feedback onset (*Miltner et al., 1997*; *Figure 4A*). It is commonly thought to index neural encoding of RPE (*Holroyd and Coles, 2002*): Its amplitude increases with the degree to which an outcome is worse than expected and, conversely, decreases to the extent that outcomes are better than expected (*Hajcak et al., 2006*; *Holroyd et al., 2006*; *Walsh and Anderson, 2012*; *Holroyd et al., 2003*; *Sambrook and Goslin, 2015*). Its amplitude thus decreases with increasing reward magnitude (*Frömer et al., 2016a*) and reward expectancy (*Lohse et al., 2020*). However, it is unknown whether reward prediction errors signaled by the FRN contrast current feedback with predictions based only on previous (external) feedback, or whether they might incorporate ongoing (internal) performance monitoring. Based on our overarching hypothesis, we predicted that FRN amplitude would scale with our estimate of RPE, which quantifies the degree to which actual feedback was 'better' than the feedback predicted after action execution, causing more negative RPEs to produce larger FRN amplitudes (*Figure 4B*). A key alternative option is that the FRN indexes the magnitude of an error irrespective of the participant's post-response outcome prediction (e.g. with large FRN to feedback indicating a large error, even when the participant knows to have committed this error) (*Pfabigan et al., 2015*; *Sambrook and Goslin, 2014*; *Talmi et al., 2013*). Note that the prediction errors experienced by most error-driven learning models would fall into this alternative category, as they would reflect the error magnitude minus some long-term expectation of that magnitude, but not update these expectations after action execution. Thus, to test whether RPE explains variation in FRN above and beyond Error

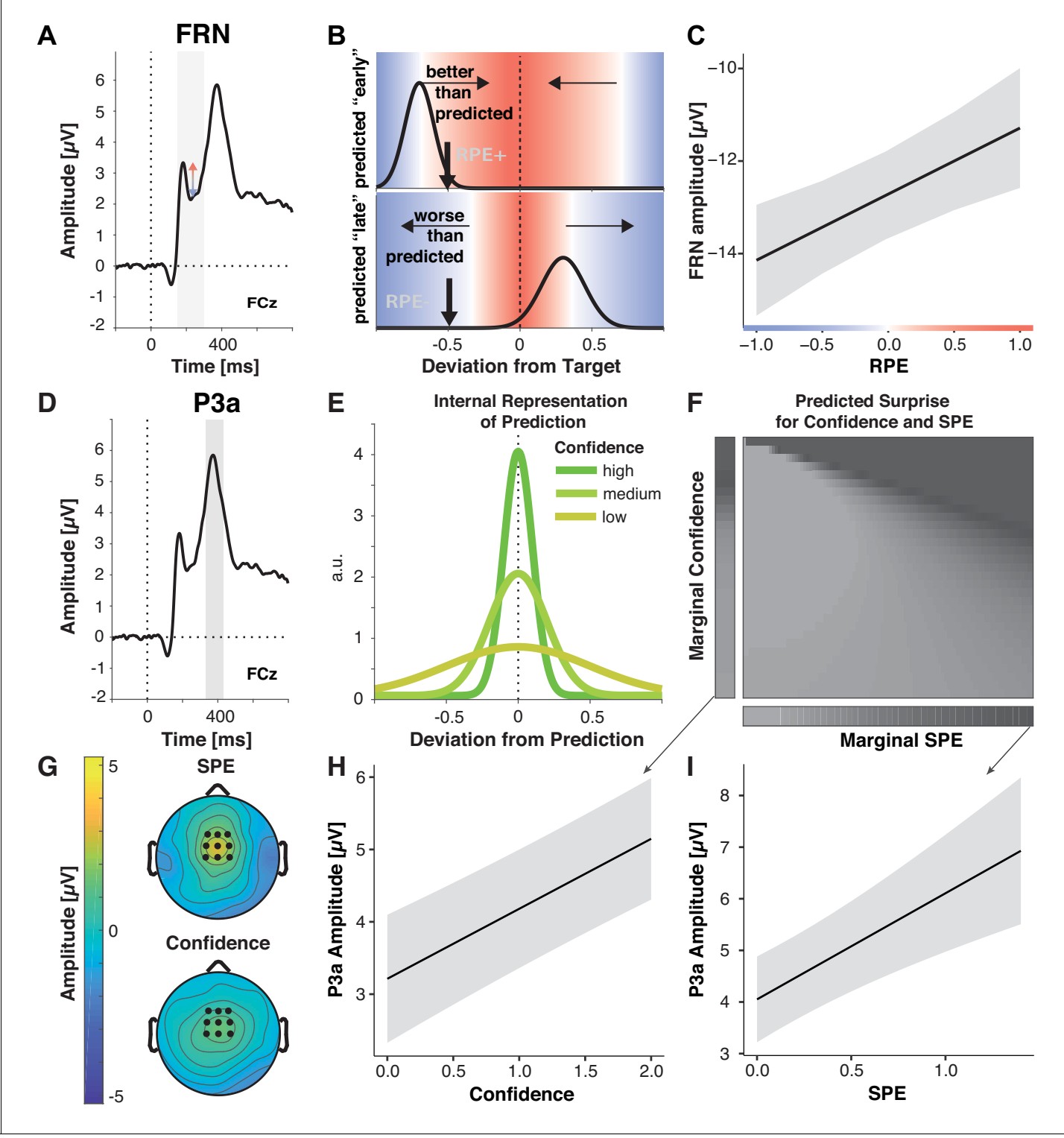

**Figure 4.** Multiple prediction errors in feedback processing. (**A-C**) FRN amplitude is sensitive to predicted error magnitude. (**A**) FRN, grand mean, the shaded area marks the time interval for peak-to-peak detection of FRN. Negative peaks between 200 and 300 ms post feedback were quantified relative to positive peaks in the preceding 100 ms time window. (**B**) Expected change in FRN amplitude as a function of RPE (color) for two predictions (black curves represent schematized predictive distributions around the reported prediction for a given confidence), one too early (top: high confidence in a low reward prediction) and one too late (bottom: low confidence in a higher reward prediction). Vertical black arrows mark a sample outcome (deviation from the target; abscissa) resulting in different RPE/expected changes in FRN amplitude for the two predictions, indicated by shades. Blue

*Figure 4 continued on next page*

*Figure 4 continued*

shades indicate negative RPEs/larger FRN, red shades indicate positive RPEs/smaller FRN and gray denotes zero. Note that these are mirrored at the goal for any predictions, and that the likelihood of the actual outcome given the prediction (y-axis) does not affect RPE. In the absence of a prediction or a predicted error of zero, FRN amplitude should increase with the deviation from the target (abscissa). (C) LMM-estimated effects of RPE on peak-to-peak FRN amplitude visualized with the effects package; shaded error bars represent 95% confidence intervals. (D– I) P3a amplitude is sensitive to SPE and Confidence. (D) Grand mean ERP with the time-window for quantification of P3a, 330–430 ms, highlighted. (E) Hypothetical internal representation of predictions. Curves represent schematized predictive distributions around the reported prediction (zero on abscissa). Confidence is represented by the width of the distributions. (F) Predictions for SPE (x-axis) and Confidence (y-axis) effects on surprise as estimated with Shannon information (darker shades signify larger surprise) for varying Confidence and SPE (center). The margins visualize the predicted main effects for Confidence (left) and SPE (bottom). (G) P3a LMM fixed effect topographies for SPE, and Confidence. (H–I) LMM-estimated effects on P3a amplitude visualized with the effects package; shaded areas in (H) (SPE) and (I) (confidence) represent 95% confidence intervals.

Magnitude, and to control for other factors, we included Error Magnitude, SPE, Confidence, and Block in the model (*Table 4*).

As predicted, FRN amplitude decreased with more positive-going RPEs ($b$ = 1.43, p<0.001, *Figure 4C*), extending previous work that investigated prediction errors as a function of reward magnitude and frequency (*Holroyd and Coles, 2002*; *Sambrook and Goslin, 2015*). In contrast, error magnitude and SPE did not significantly affect FRN amplitude, suggesting in the case of the error magnitude that when errors can be accounted for by faulty execution, they do not drive internal reward prediction error signals. We found no other reliable effects and when including interaction terms, they were neither significant, nor supported by model selection ($\Delta X^2(10)$=10.98, p=0.359, $AIC_{reduced-full}$ = −9, $BIC_{reduced-full}$ = −81). We conclude that FRN amplitude reflects the degree to which feedback is better than predicted, and critically, that the outcome predictions incorporate information about likely execution errors.

## Sensory prediction error and confidence modulate P3a

The frontocentral P3a is a surprise-sensitive positive-going deflection between 250 and 500 ms following stimulus onset (*Figure 3E*; *Polich, 2007*). Its functional significance can be summarized as signaling the recruitment of attention for action to surprising and motivationally relevant stimuli (*Polich, 2007*; *Nieuwenhuis et al., 2011*). P3a has been shown to increase with larger prediction errors in probabilistic learning tasks (*Fischer and Ullsperger, 2013*), higher goal-relevance in a go/no-go task (*Walentowska et al., 2016*), with increasing processing demands (*Frömer et al., 2016b*), and with meta-memory mismatch (feedback about incorrect responses given with high confidence [*Butterfield and Mangels, 2003*]).

**Table 4.** LMM statistics of learning effects on FRN.

| Predictors | Peak-to-Peak FRN amplitude | | | | |
|---|---|---|---|---|---|
| | *Estimates* | *SE* | *CI* | *t* | *p* |
| Intercept | −12.67 | 0.49 | −13.62 – −11.71 | −26.03 | **2.322e-149** |
| Confidence | −0.19 | 0.15 | −0.49–0.11 | −1.25 | 2.126e-01 |
| Reward prediction error | 1.43 | 0.41 | 0.62–2.24 | 3.47 | **5.302e-04** |
| Sensory prediction error | −0.67 | 0.42 | −1.49–0.15 | −1.61 | 1.078e-01 |
| Error magnitude | 0.51 | 0.55 | −0.57–1.58 | 0.92 | 3.553e-01 |
| Block | −0.15 | 0.11 | −0.36–0.06 | −1.43 | 1.513e-01 |
| **Random effects** | | | **Model Parameters** | | |
| Residuals | 27.69 | | $N_{vpn}$ | 40 | |
| Intercept | 9.23 | | Observations | 9678 | |
| Error magnitude | 2.24 | | log-Likelihood | −29908.910 | |
| Block | 0.22 | | Deviance | 59817.821 | |

Formula: FRN ~ Confidence + RPE+SPE + EM+Block + (EM +Block|participant).

Surprise can be quantified using Shannon Information, which reflects the amount of information provided by an outcome given a probability distribution over outcomes (*O'Reilly et al., 2013*). As seen in *Figure 4F*, this measure scales with increasing confidence, as well as SPE, that is, increasing deviations between predicted and actual outcome (margins). To generate these predictions, we computed the Shannon Information for a range of outcomes given a range of predictive distributions with varying precision, assuming that confidence reflects the precision of a distribution of predicted outcomes (*Figure 4E*). Thus, P3a amplitude should scale with both SPE and Confidence. We tested our predictions by examining whether P3a was modulated by SPE, and Confidence, in a model that also included Error Magnitude, RPE, and Block as control variables.

As predicted, our analyses showed that P3a amplitude significantly increased with increasing SPEs, in line with the idea of stronger violations of expectations by less accurately predicted outcomes (*Figure 4G and I*, *Table 5*), and with increasing Confidence (*Figure 4G and H*). Our Shannon Information simulation also predicts a small interaction between SPE and Confidence (see slight diagonal component in *Figure 4F*). However, when including the interaction term it was not significant and did not improve model fit, $\Delta X^2(10)=10.36$, p=0.410 (AIC $_{reduced-full}$ = −10, BIC $_{reduced-full}$ = −81), suggesting that any such effect was minimal.

In addition, P3a amplitude decreased across blocks, perhaps reflecting decreased motivational relevance of feedback as participants improved their performance and predictions (*Walentowska et al., 2016*; *Severo et al., 2020*). We also observed a significant decrease of P3a with increasing Error Magnitude and larger P3a amplitudes for more negative reward prediction errors. However, these effects showed more posterior scalp distributions than those of SPE and confidence. As P3a temporally overlaps with the more posteriorly distributed P3b, these effects are likely a spillover of the P3b. Hence, we discuss them below. Taken together our results support our hypothesis that predictions and confidence shape feedback processing at the level of the P3a.

## Prediction errors, objective errors, and confidence converge in the P3b

Our overarching hypothesis and model predict that outcome predictions and confidence should affect the degree to which feedback is used for future behavioral adaptation. The parietally distributed P3b scales with learning from feedback (*Ullsperger et al., 2014a*; *Fischer and Ullsperger, 2013*; *Yeung and Sanfey, 2004*; *Sailer et al., 2010*; *Chase et al., 2011*) and predicts subsequent behavioral adaptation (*Fischer and Ullsperger, 2013*; *Chase et al., 2011*). P3b amplitude has been found to increase with feedback salience (reward magnitude irrespective of valence; *Yeung and Sanfey, 2004*), behavioral relevance (choice vs. no choice; *Yeung et al., 2005*), with more negative-going RPE (*Ullsperger et al., 2014a*; *Fischer and Ullsperger, 2014*), but also with better outcomes in more complex tasks (*Pfabigan et al., 2014*).

**Table 5.** LMM statistics of learning effects on P3a.

| Predictors | P3a Amplitude | | | | |
| | Estimates | SE | CI | t | p |
|---|---|---|---|---|---|
| Intercept | 4.10 | 0.42 | 3.28–4.93 | 9.79 | 1.293e-22 |
| Confidence | 0.97 | 0.14 | 0.70–1.24 | 6.96 | 3.338e-12 |
| Block | −0.91 | 0.07 | −1.05 – −0.77 | −12.93 | 3.201e-38 |
| Sensory prediction error | 2.06 | 0..48 | 1.11–3.00 | 4..27 | 1.969e-05 |
| Reward prediction error | −0.75 | 0.38 | −1.49 – −0..01 | −1.98 | 4.794e-02 |
| Error magnitude | −1..95 | 0..44 | −2.81 – −1..09 | −4.43 | 9.512e-06 |
| **Random effects** | | | **Model Parameters** | | |
| Residuals | 22.98 | | N | 40 | |
| Intercept | 6.83 | | Observations | 9678 | |
| SPE | 3.02 | | log-Likelihood | −28997.990 | |
| | | | Deviance | 57995.981 | |

Formula: P3a ~ Confidence + Block +SPE + RPE+EM + (SPE|participant).

Consistent with their necessity for governing behavioral adaptation, P3b was sensitive to participants' outcome predictions (*Table 6*). P3b amplitude increased with increasing SPE (*Figure 5B,E*), indicating that participants extracted more information from the feedback stimulus when outcomes were less expected. As for P3a, this SPE effect decreased across blocks, and so did overall P3b amplitude, suggesting that participants made less use of the feedback as they improved on the task (*Fischer and Ullsperger, 2013*).

P3b amplitude also increased with negative-going RPE (*Figure 5B,C*), hence, for worse-than-expected outcomes, replicating previous work (*Ullsperger et al., 2014a*; *Fischer and Ullsperger, 2014*). This RPE effect was significantly modulated by Confidence and Block, indicating that the main effect needs to be interpreted with caution, and the relationship between P3b and RPE is more nuanced than previous literature suggested. As shown in *Figure 5D* in the first block, P3b amplitude was highest for large negative RPE and *high* Confidence, whereas in the last block it was highest for large negative RPE and *low* Confidence (see below for follow-up analyses).

In line with previous findings (*Pfabigan et al., 2014*; *Ernst and Steinhauser, 2018*), we further observed significant increases of P3b amplitude with decreasing Error Magnitude, thus, with better outcomes (*Figure 5B,F*, *Table 5*). We found no further significant interactions, and excluding the non-significant interaction terms from the full model did not significantly diminish goodness of fit, $\Delta X^2(5)=10.443$, p=0.064 (AIC$_{reduced-full}$ = 0; BIC$_{reduced-full}$ = −35).

Our hypothesis states that the degree to which people rely on their predictions when learning from feedback should vary with their confidence in those predictions. In the analysis above, we observed such an interaction with confidence only for RPE (and Block). RPE is derived from the contrast between Error Magnitude and Predicted Error Magnitude, and changes in either variable or their weighting could drive the observed interaction. To better understand this interaction and test explicitly whether confidence regulates the impact of predictions, we therefore ran complementary analyses where instead of RPE we included Predicted Error Magnitude (*Table 7*). Confirming its relevance for the earlier interaction involving RPE, Predicted Error Magnitude indeed interacted with Confidence and Block. Consistent with confidence-based regulation of learning, a follow-up analysis showed that in the first block, P3b significantly increased with higher Confidence, and importantly decreased significantly more with increasing Predicted Error Magnitude as Confidence increased

**Table 6.** LMM statistics of learning effects on P3b.

| Predictors | P3b Amplitude | | | | |
| --- | --- | --- | --- | --- | --- |
| | Estimates | SE | CI | t | p |
| Intercept | 4.12 | 0.29 | 3.55–4.70 | 14.12 | **2.937e-45** |
| Block | −0.48 | 0.09 | −0.66 − −0.30 | −5.20 | **2.037e-07** |
| Confidence | 0.08 | 0.20 | −0.31–0.48 | 0.42 | 6.740e-01 |
| Reward prediction error | −1.12 | 0.46 | −2.03 − −0.22 | −2.43 | **1.493e-02** |
| Sensory prediction error | 1.75 | 0.47 | 0.84–2.66 | 3.76 | **1.691e-04** |
| Error magnitude | −2.35 | 0.46 | −3.24 − −1.45 | −5.14 | **2.743e-07** |
| Confidence: Reward prediction error | −0.51 | 0.55 | −1.60–0.57 | −0.92 | 3.556e-01 |
| Block: Confidence | 0.07 | 0.18 | −0.28–0.43 | 0.41 | 6.823e-01 |
| Block: Reward prediction error | −0.52 | 0.44 | −1.39–0.34 | −1.19 | 2.359e-01 |
| Block: Sensory prediction error | −0.98 | 0.46 | −1.88 − −0.07 | −2.12 | **3.405e-02** |
| Block: Confidence: Reward prediction error | 2.22 | 0.72 | 0.81–3.64 | 3.08 | **2.057e-03** |
| **Random effects** | | | | | |
| Residuals | 23.95 | | N | 40 | |
| Intercept | 3.17 | | Observations | 9678 | |
| Sensory Prediction Error | 2.16 | | log-Likelihood | −29197.980 | |
| Reward prediction error | 1.67 | | Deviance | 58395.960 | |
| Confidence | 0.63 | | | | |

Formula: P3b ~ Block*(Confidence*RPE +SPE)+Error Magnitude + (SPE +RPE + Confidence|participant); Note: ':" indicates interactions.

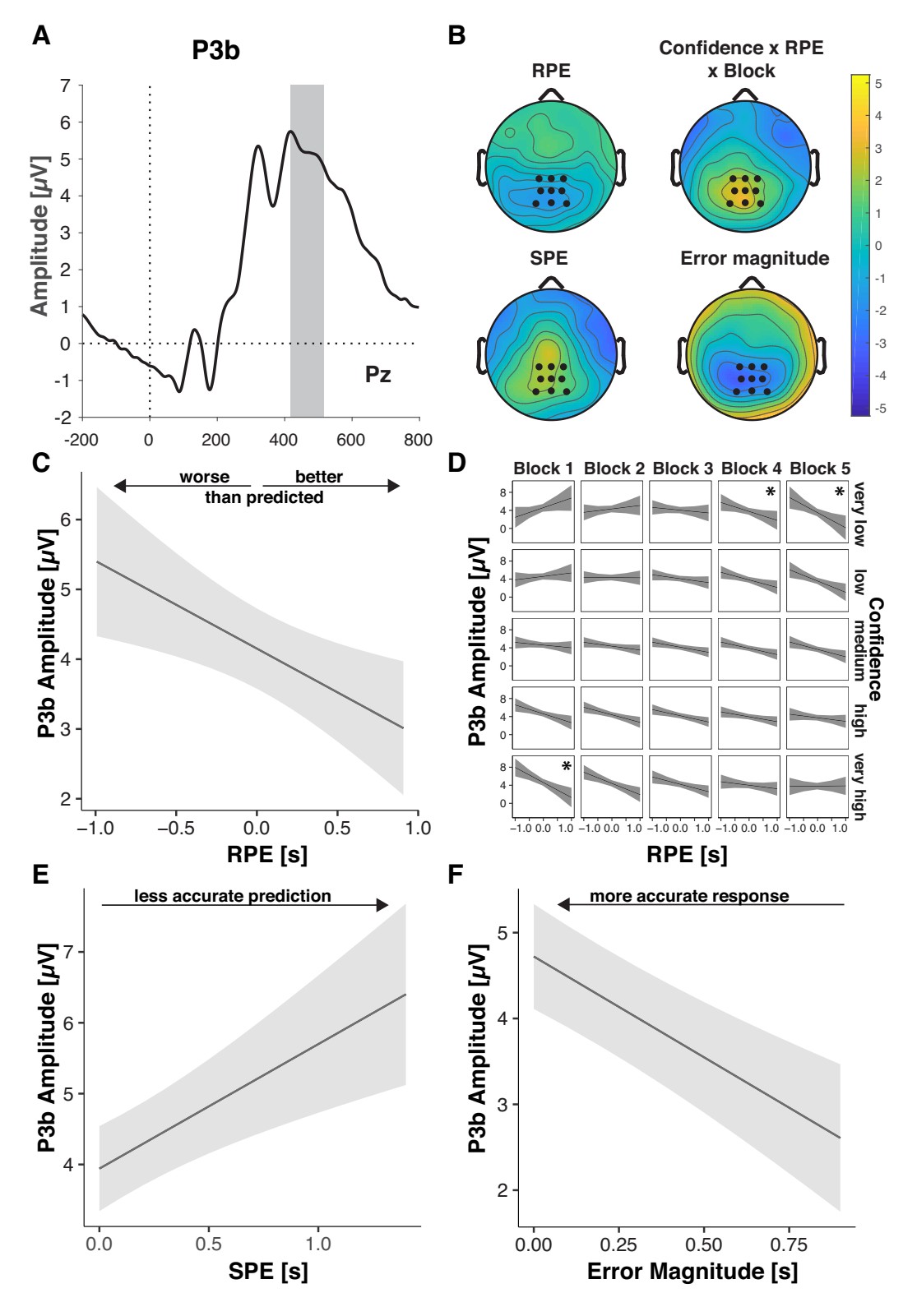

**Figure 5.** Performance-relevant information converges in the P3b. (A) Grand average ERP waveform at Pz with the time window for quantification, 416–516 ms, highlighted. (B) Effect topographies as predicted by LMMs for RPE, error magnitude, SPE and the RPE by Confidence by Block interaction. (C–F) LMM-estimated effects on P3b amplitude visualized with the effects package in R; shaded areas represent 95% confidence intervals. (C.) RPE. Note
*Figure 5 continued on next page*

*Figure 5 continued*

the interaction effects with Block and Confidence (D), that modulate the main effect (D) Three-way interaction of RPE, Confidence and Block. Asterisks denote significant RPE slopes within cells. (E) P3b amplitude as a function of SPE. (F) P3b amplitude as a function of Error Magnitude.

The online version of this article includes the following figure supplement(s) for figure 5:

**Figure supplement 1.** P3b to feedback modulates error-related adjustments on the subsequent trial.

(*Supplementary file 8*). Main effects of Predicted Error Magnitude emerged only in the late blocks when participants were overall more confident.

Hence, our P3b findings indicate that early on in learning, when little is known about the task, participants learn more and discount their predictions more when they have high confidence in those predictions. In later trials however, when confidence is higher overall, participants discount their predicted errors even when confidence is relatively lower.

We next explored whether P3b amplitude is associated with trial-by-trial adjustments. To that aim, we computed the improvement on trial n as the difference between the error on trial n and the error on trial n-1. Time-estimation responses are noisy, and thus provide only a coarse trial-by-trial indicator of learning. Consistent with regression to the mean, where larger errors are more likely followed by smaller errors, improvements increased with the magnitude of the error on the previous trial (b = 0.85, p<0.001, *Supplementary file 9*). We find, however, that this effect varies across blocks (b = 0.13, p<0.001), and is least pronounced in the first block when most learning takes place (Block 1: b = 0.66, p<0.001; Block 2–5: b >= 0.90, p<0.001, *Supplementary file 10*). We thus next tested whether P3b on trial n-1 mediates the relationship between error magnitude on trial n-1 and the improvement on the current trial, leading to stronger improvements following a given error, particularly in the first block when most learning takes place and performance is less determined by previous error alone. Indeed, we found a significant three-way interaction between previous error magnitude, previous P3b amplitude and Block (b = - 0.03, p=0.031, *Supplementary file 9*, *Figure 5—figure supplement 1*) on improvement. A follow-up analysis confirmed that P3b mediated the relationship between previous error magnitude and improvement in the first block (b = 0.06, p<0.001, *Supplementary file 10*). This interaction was not significant within any of the remaining

**Table 7.** LMM statistics of confidence weighted predicted error discounting on P3b.

| | **P3b Amplitude** | | | | |
|---|---|---|---|---|---|
| *Predictors* | *Estimates* | *SE* | *CI* | *t* | *p* |
| Intercept | 4.26 | 0.30 | 3.68–4.85 | 14.22 | **7.239e-46** |
| Confidence | 0.31 | 0.22 | −0.12–0.75 | 1.41 | 1.595e-01 |
| Predicted error magnitude | −0.83 | 0.46 | −1.74–0.07 | −1.80 | 7.133e-02 |
| Block | −0.32 | 0.11 | −0.52 − −0.11 | −2.98 | **2.860e-03** |
| Error magnitude | −1.06 | 0.49 | −2.03 − −0.09 | −2.13 | **3.277e-02** |
| Sensory prediction error | 1.49 | 0.40 | 0.71–2.28 | 3.72 | **1.992e-04** |
| Confidence: Predicted error magnitude | −0.98 | 0.69 | −2.34–0.38 | −1.41 | 1.582e-01 |
| Confidence: Block | −0.50 | 0.20 | −0.90 − −0.11 | −2.50 | **1.249e-02** |
| Predicted Error magnitude: Block | −1.12 | 0.56 | −2.22 − −0.02 | −2.00 | **4.540e-02** |
| Confidence: Predicted error magnitude: Block | 3.12 | 0.84 | 1.47–4.78 | 3.70 | **2.141e-04** |
| **Random effects** | | | **Model Parameters** | | |
| Residuals | 23.98 | | N | 40 | |
| Intercept | 3.30 | | Observations | 9678 | |
| Error magnitude | 3.43 | | log-Likelihood | −29201.951 | |
| Confidence | 0.72 | | Deviance | 58403.902 | |

Formula: P3b ~ Block*(Confidence*Predicted Error Magnitude +SPE)+Error Magnitude + (Error Magnitude +Confidence|participant); Note: ':" indicates interactions.

blocks. While intriguing and in line with previous work linking P3b to trial-by-trial adjustments in behavior, these results should be interpreted with a degree of caution given that the present task is not optimized to test for trial-to-trial adjustments in behavior.

Taken together, our ERP findings support our main hypothesis that individuals take their internal evaluations into account when processing feedback, such that distinct ERP components reflect different aspects of internal evaluations rather than just signaling objective error.

## Discussion

The present study explored the hypothesis that learning from feedback depends on internal performance evaluations as reflected in outcome predictions and confidence. Comparing different Bayesian agents with varying insights into their trial-by-trial performance, we show that performance monitoring provides an advantage in learning, as long as agents' confidence is accurately calibrated. To test our hypothesis empirically, we collected participants' trial-wise outcome predictions and confidence judgments in a time-estimation task prior to receiving feedback, while recording EEG. Like the simulations from the Bayesian learner with performance monitoring, our empirical results show that trial-by-trial confidence tracks the precision of outcome predictions, and individuals with better coupling between confidence and the precision of their predictions (confidence calibration) showed greater improvements in performance over the course of the experiment. Moreover, participants' subjective predictions, as well as their confidence in those predictions, influenced feedback processing as revealed by feedback-related potentials.

Our study builds on an extensive body of work on performance monitoring, proposing that deviations from performance goals are continuously monitored, and expectations are updated as soon as novel information becomes available (*Holroyd and Coles, 2002*; *Ullsperger et al., 2014b*). Hence, performance monitoring at later stages should depend on performance monitoring at earlier stages (*Holroyd and Coles, 2002*). Specifically, learning from feedback should critically depend on internal performance monitoring. Our results extend previous work demonstrating a shift from feedback-based to response-based evaluations as learning progresses (*Bellebaum and Colosio, 2014*; *Bultena et al., 2017*): They show that performance monitoring and learning from feedback are not mutually exclusive modes of performance evaluation; instead, they operate in concert, with confidence in response-based outcome predictions determining the degree to which this information is relied on.

Participants' behavior displayed hallmarks of error monitoring (*Kononowicz et al., 2019*; *Akdoğan and Balcı, 2017*; *Kononowicz and van Wassenhove, 2019*), such that outcome predictions tracked factual errors in both direction and magnitude. Crucially, extending those previous findings, our empirical results align with unique predictions based on our hypothesis: Confidence reflected the precision of participants' outcome predictions, and participants with superior calibration of their confidence judgments to the accuracy of their predictions learned better than those with poorer calibration. This latter finding is notable given that overall confidence calibration was similar for participants with different performance quality (error magnitude, response variance). Therefore, the empirical confidence calibration effect on learning is unlikely to be a consequence of better overall ability as described in the 'unskilled and unaware effect' (*Kruger and Dunning, 1999*) or of the dependence of confidence calibration (or metacognitive sensitivity) on performance (*Fleming and Lau, 2014*). Instead, the finding supports our hypothesis that confidence supports learning via optimized feedback processing.

Our simulations and ERP data reveal two critical mechanisms through which performance monitoring may impact learning from feedback: modulation of surprise and reduction of uncertainty via credit assignment. The main impact of error monitoring is to *reduce* the surprise about outcomes. All else being equal, a given outcome is less surprising the better it matches the predicted outcome. Consistent with discounting of predicted deviations from the goal, we found that participants' trial-by-trial subjective outcome predictions consistently modulated feedback-based evaluation reflected in ERPs as evidenced by prediction error effects. Participants' response-based outcome predictions were reflected in the amplitudes of FRN reward prediction error signals (*Holroyd and Coles, 2002*; *Walsh and Anderson, 2012*; *Sambrook and Goslin, 2015*; *Correa et al., 2018*), of P3a surprise signals, as well as the P3b signals combining information about reward prediction error and surprise. In our computational models, reducing surprise by taking response-based outcome predictions into

account led to more accurate updating of internal representations supporting action selection and thus superior learning.

Learning – in our computational model and in our participants - was further supported by the adaptive regulation of uncertainty-driven updating via confidence. Specifically, as deviations from the goal were predicted with higher confidence, these more precise outcome predictions *enhanced* the surprise elicited by a given prediction error. This mechanism implemented in our model is mirrored in participants' increased P3a amplitudes for higher confidence, and further reflected in confidence-weighted impacts of predicted error magnitude on P3b, as well as larger P3b amplitudes for higher confidence in the first block when most learning took place. Thus, a notable finding revealed by our simulations and empirical data is that, counterintuitively, agents and participants learned *more* from feedback when confidence in their predictions had been high.

Although FRN amplitude was not modulated by confidence, we found that P3a increased with confidence, as predicted by uncertainty-driven changes in surprise. Our results align with previous findings of larger P3a amplitude for metacognitive mismatch (*Butterfield and Mangels, 2003*) and offer a computational mechanism underlying previous theorizing that feedback about errors committed with high confidence attracts more attention, and therefore leads to hypercorrection (*Butterfield and Metcalfe, 2006*; *Butterfield and Metcalfe, 2001*). We also found that confidence modulated the degree to which predicted error magnitude reduced P3b amplitude, such that in initial blocks, where most learning took place, predicted error magnitude effects were amplified for higher confidence, whereas this effect diminished in later blocks, where predicted error magnitude effects were present also for low confidence (and performance and prediction errors were attenuated when confidence was high). This shift is intriguing and may indicate a functional change in feedback use as certainty in the response-outcome mapping increases and less about this mapping is learned from feedback, but the effect was not directly predicted and therefore warrants further research and replication.

Confidence has typically been studied in two-alternative choice tasks, and only rarely in relation to continuous outcomes (*Meyniel et al., 2015*; *Meyniel and Dehaene, 2017*; *Boldt et al., 2019*; *Lebreton et al., 2015*; *Nassar et al., 2012*; *Arbuzova, 2020*). By reconceptualizing error detection as outcome prediction, our results shed new light on the well-supported claim that error monitoring and confidence are tightly intertwined (*Boldt and Yeung, 2015*; *Yeung and Summerfield, 2012*; *Charles and Yeung, 2019*; *Desender et al., 2018b*; *Desender et al., 2019*) and forge valuable links between research on performance monitoring (*Ullsperger et al., 2014a*; *Holroyd and Coles, 2002*; *Ullsperger et al., 2014b*) and on learning under uncertainty (*McGuire et al., 2014*; *Behrens et al., 2007*; *O'Reilly et al., 2013*; *Nassar et al., 2019*). In doing so, our results provide further evidence to the growing literature on the role of confidence in learning and behavioral adaptation (*Meyniel and Dehaene, 2017*; *Desender et al., 2018a*; *Boldt et al., 2019*; *Colizoli et al., 2018*).

While we captured the main effects of interest with our Bayesian model and our key behavioral results are in line with our overarching hypothesis, our behavioral findings also reveal other aspects of learning that remain to be followed up on. Unlike our Bayesian agents, participants exhibited signatures of learning not only at the level of first order performance, but also at the level of performance monitoring. The precision of their outcome predictions increased as learning progressed, as did confidence. Identifying the mechanisms that drive this metacognitive learning, that is, whether changes in confidence follow the uncertainty in the internal model or reflect refinement of the confidence calibration to the efference copy noise, is an exciting question for future work.

Anyone who tried to learn a sport can relate to the intuition that just because you find out what you did was wrong doesn't mean you know how to do it right. Our task also evokes this so-called distal problem, which refers to the difficulty of translating distal sensory outcomes of responses (e.g. the location of a red dot on a feedback scale) to required proximal movement parameter changes (changes in the timing of the response). Indeed, when practicing complex motor tasks, individuals prefer and learn better from feedback following successful trials compared to error trials (*Chiviacowsky and Wulf, 2007*; *Chiviacowsky and Wulf, 2002*; *Chiviacowsky and Wulf, 2005*). In line with the notion that in motor learning feedback about success is more informative than feedback about failure, we, like others in the time estimation task (*Pfabigan et al., 2014*; *Ernst and Steinhauser, 2018*), observed increasing P3b amplitude after feedback about more accurate performance (i.e. for smaller error magnitude), in addition to prediction error effects.

In our study, the P3b component, previously shown to scale with the amount of information provided by a stimulus (*Cockburn and Holroyd, 2018*) and the amount of learning from a stimulus (*Fischer and Ullsperger, 2013*), was sensitive to both RPE and SPE, indicating that multiple learning mechanisms may act in parallel, supported by different aspects of feedback. Our findings resonate with recent work in rodents showing that prediction error signals encode multiple features of outcomes (*Langdon et al., 2018*), and are based on distributed representations of predictions (*Dabney et al., 2020*). This encoding of multiple features of outcomes, like the uncertainty in predictions, may help credit assignment and support learning at multiple levels. It is still unclear to what degree different learning mechanisms – error-based, model-based, reinforcement learning – contribute to motor learning (*Wolpert and Flanagan, 2016*). Further research is needed to identify whether the same or different learning mechanisms operate across levels, for example, via hierarchical reinforcement learning (*Holroyd and Yeung, 2012*; *Lieder et al., 2018*), and how learning interacts between levels.

Taken together, our findings provide evidence that feedback evaluation is fundamentally affected by an individual's internal representations of their own performance at the time of feedback. These internal representations in turn influence how people learn and thus which beliefs they will have and which actions they will take, driving what internal and external states they will encounter in the future. The present study is a first step toward elucidating this recursive process of performance optimization via internal performance monitoring and monitoring of external task outcomes.

## Materials and methods

### Task variables

- $t$ denotes the target interval, which was set to $t := 19$, (this simulation parameter choice was necessarily somewhat arbitrary, and choosing a different parameter does not change the model's behavior)
- $s$ denotes the feedback scale, which was set to $s := 90$,
- $r$ denotes the model's or participant's response,
- $f$ denotes the feedback in the task, which was defined as

$$f := (r - t)s \tag{1}$$

### Computational model

The Bayesian learner with performance monitoring attempted to sequentially infer the target interval $t$ and feedback scale $s$ (defining how the magnitude of a given response error translates to the magnitude of the error displayed on the visual feedback scale) over multiple trials, based on its intended response $i$, an efference copy $c$ of its executed response and feedback $f$ indicating the magnitude and direction of its timing errors. On each trial the model computed its intended response based on the inferred target interval. During learning, the model faced several obstacles including (1) the initially unknown scale of the feedback, making it difficult to judge whether feedback indicates small or large timing errors, (2) response noise, which offsets executed responses from intended ones, and (3) efference copy noise, which makes the efference copy unreliable to a degree that varies from trial to trial. Formally, the Bayesian learner with performance monitoring is represented by the following variables:

- $p(t)U(t; [0, 100])$ denotes the model's prior distribution over the target interval $t$, as a uniform distribution (denoted by U throughout) of over possible values of t within the range 0 to 100.
- $p(s)U(s; [0.1, 100])$ denotes the model's prior distribution over the feedback scale $s$, as a uniform distribution over possible values of s within a range of 0.1 to 100.
- $p^{\sigma_r^2}(r|i)N(r; i, \sigma_r^2)$ denotes the model's response distribution (N denoting normal distributions throughout), where $i$ denotes the model's intended response, which corresponds to the expected target interval $i := E_{p(t)} = \sum_t p(t)t$, and $\sigma_r^2$ denotes the response noise, which was set to $\sigma_r := 10$ in terms of the standard deviation
- $p^{\sigma_c^2}(c|r)N(c; r, \sigma_c^2)$ denotes the model's efference-copy distribution with efference-copy noise (we simulated three levels: low, medium and high) expressed as standard

deviation $\sigma_c \in \{5, 10, 20\}$, where $p(\sigma_c)Cat(\sigma_c, 1/3)$. Here we assumed that the model was aware of its trial-by-trial efference copy noise. That is, from the perspective of the Bayesian learner with performance monitoring, efference-copy noise was not a random variable.

## Intended response

During the response phase in the task, the model first computed its intended response $i$ and then sampled its actual response $r$ from the Gaussian response distribution. We assumed that the model's internal response monitoring system subsequently generated the noisy efference copy $c$.

## Learning

Based on the definition of the task and the Bayesian learner with performance monitoring, the joint distribution over the variables of interest during a trial of the task is given by

$$p^{i,\sigma_c^2,\sigma_r^2}(t,s,f,r,c) := p(f|r,t,s)p^{\sigma_c^2}(c|r)p^{i,\sigma_r^2}(r)p(t,s) \tag{2}$$

To infer the target interval $t$ and the feedback scale $s$, we can evaluate the posterior distribution conditional on the efference copy $c$ and feedback $f$ and given the intended response $i$, response noise $\sigma_r^2$ and efference copy noise $\sigma_c^2$ according to Bayes' rule:

$$p^{i,\sigma_c^2,\sigma_r^2}(t,s|c,f)$$
$$\propto \int p^{i,\sigma_c^2,\sigma_r^2}\left(t,s|c,f,r\right)dr \tag{3}$$
$$\propto \int p\left(f|r,t,s\right)p^{\sigma_c^2}\left(c|r\right)p^{i,\sigma_r^2}(r)p(t,s)dr$$

Note that we assumed that the Bayesian learner with performance monitoring was aware how feedback $f$ was generated in the task (*Equation 1*), that is, conditional on the response $r$, target interval $t$ and feedback scale $s$, the model was able to exactly compute the probability of the feedback according to

$$p(f|r,t,s) = \begin{cases} 1, iff = (r-t)s \\ 0, else \end{cases} \tag{4}$$

We approximated inference using a grid over the target interval $t \in [0, 100]$ and feedback scale $s \in [0.1, 100]$. The model first computed the probability of the currently received feedback. Although it was aware how feedback was generated in the task, it suffered from uncertainty over its response due to noise in the efference copy. For each $s$ and $t$ on the grid, the model evaluated the Gaussian distribution

$$p^{i,\sigma_c^2,\sigma_r^2}(f|c,t,s) := N\left(f;(m-t)^T s, vA\right) \tag{5}$$

where A was a $100 \times 50$ matrix containing the grid values of $t$ and $s$.

$$v = \frac{1}{\frac{1}{\sigma_c^2} + \frac{1}{\sigma_r^2}} \tag{6}$$

denotes the expected variance in feedback under consideration of both efference-copy noise $\sigma_c^2$ and response noise $\sigma_r^2$ and

$$m = v\frac{1}{\sigma_c^2}c + v\frac{1}{\sigma_r^2}i \tag{7}$$

denotes the expected feedback under the additional consideration of efference copy $c$ and intended response $i$. When computing the probability of the feedback, our model thus took into account the efference copy $c$ and the response it intended to produce $i$, which were weighted according to their respective reliabilities.

Second, the model multiplied the computed probability of the observed feedback with the prior over the scale and target response, that is

$$p^{i,\sigma_c^2,\sigma_r^2}(t,s|c,f) \propto N(f;(m-t)^T s, vA)p(s,t) \tag{8}$$

In the grid approximation, the model started with a uniform prior on the joint distribution over $t$ and $s$ and was applied recursively, such that the posterior joint distribution on $t$ and $s$ for each trial served as the prior distribution for the subsequent one.

### Outcome prediction

On each trial, the model reports an outcome prediction $po$ and the confidence in this prediction, which we refer to as $co$. The outcome prediction maps the discrepancy between the intended response and the efference copy onto the feedback scale given the best guess of the current $t$ and $s$.

$$po = (c-i)\sum_s p(s)s \tag{9}$$

Note that this outcome prediction is different from the mean of the uncertainty-weighted expectancy distribution defined in *Equation 7*, in that it does not take uncertainty into account, but reflects the expectation given the efference copy alone. The inverse uncertainty $\sigma_c^2$ of the efference copy, as described below, is translated into the agent's confidence report.

### Confidence calibration

Confidence calibration $cc \in \{0, 0.75, 1\}$ denotes the probability that the agent assumes the correct efference copy variance $\sigma_c^2$ for learning (c.f. *Equations 6,7*). $cc = 1$ indicates that the subjectively assumed efference-copy precision is equal to the true precision and $cc = 0$, in contrast, indicates that the assumed precision of the efference copy is different from the true one. That is, in the case of $cc = 0.75$, the agent more likely assumes the true precision of the efference copy but sometimes fails to take it accurately into account during learning. As shown in *Figure 1*, we simulated behavior of three agents that differed in their confidence calibration according to this idea.

### Confidence report

In our model, the confidence report $co \in \{3, 2, 1\}$ (we simulate three levels as for efference copy noise) reflects how certain the agent thinks its efference copy is, where 3 refers to 'completely certain', and 1 to 'not certain'. In particular,

$$co = \begin{cases} 3, & if\, \sigma_c = 5 \\ 2, & if\, \sigma_c = 10 \\ 1, & if\, \sigma_c = 20 \end{cases} \tag{10}$$

That is, the confidence report is directly related to the subjective precision of the efference copy, which, as shown above, depends to the agent's level of confidence calibration.

### Model with incomplete performance monitoring

We also applied a model that had no insight into the precision of its current predictions. The agent was thus unaware of its trial-by-trial efference-copy noise $\sigma_c$ and therefore relied on the average value of $\sigma_c$, that is $\sum \sigma_c p(\sigma_c) = 12$. As the model does not differentiate between precise and imprecise predictions but treats each prediction as average precise, it relies too much on imprecise predictions and too little on precise ones.

### Model without performance monitoring

Finally, we applied a model that was aware about its response noise but, because it completely failed to consider its efference copy, it lacked insight into its trial-by-trial performance. In this version, we had

$$v = \frac{1}{\frac{1}{\sigma_r^2}} \tag{11}$$

and $m = i$.

This model accounts for the expected variance in the feedback, but cannot differentiate between trials in which the feedback is driven more by incorrect beliefs about the target versus incorrect execution. Therefore, it adjusts its beliefs too much following feedback primarily driven by execution errors and too little following feedback primarily driven by incorrect beliefs.

## Participants

The experimental study included 40 participants (13 males) whose average age was 25.8 years ($SD$ = 4.3) and whose mean handedness score (*Oldfield, 1971*) was 63.96 ($SD$ = 52.09; i.e., most participants were right-handed). Participants gave informed consent to the experiment and were remunerated with course credits or 8 € per hour.

## Task and procedure

Participants performed an adapted time-estimation task (*Luft et al., 2014*; *Miltner et al., 1997*) that included subjective accuracy and confidence ratings (similar to *Kononowicz et al., 2019*; *Akdoğan and Balcı, 2017*; *Kononowicz and van Wassenhove, 2019*). Participants were instructed that their primary goal in this task is to learn to produce an initially unknown time interval. In addition, they were asked to predict the direction and magnitude of any errors they produced and their confidence in those predictions. The time-estimation task is well established for ERP analyses e.g. (*Luft et al., 2014*; *Miltner et al., 1997*), and has the advantages that it limits the degrees of freedom of the response, and precludes concurrent visual feedback that might affect performance evaluation. The task consisted of four parts on each trial, illustrated in *Figure 1B*. After a fixation cross lasting for a random interval of 300–900 ms, a tone (600 Hz, 200 ms duration) was presented. Participants' task was to terminate an initially unknown target interval of 1504 ms from tone onset, by pressing a response key with their left hand. We chose a supra-second duration to make the task sufficiently difficult (*Luft et al., 2014*). Following the response, a fixation cross was presented for 800 ms. Participants then estimated the accuracy of the interval they had just produced by moving an arrow on a visual analogue scale (*too short – too long*;±125 pixel, 3.15 ˚ visual angle) using a mouse cursor with their right hand. Then, on a scale of the same size, participants rated their confidence in this estimate (*not certain – fully certain*). The confidence rating was followed by a blank screen for 800 ms. Finally, participants received feedback about their performance with a red square (0.25˚ visual angle) placed on a scale identical to the accuracy estimation scale but without any labels. The placement of the square on the scale visualized error magnitude in the interval produced, with undershoots shown to the left and overshoots on the right side of the center mark, indicating the correct estimate. Feedback was presented for only 150 ms to preclude eye movements. The interval until the start of the next trial was 1500 ms.

The experiment comprised five blocks of 50 trials each, with self-paced rests between blocks. We used Presentation software (Neurobs.) for stimulus presentation, event and response logging. Visual stimuli were presented on a 4/3 17'' BenQ Monitor (resolution: 1280 × 1024, refresh rate: 60 Hz) placed at 60 cm distance from the participant. A standard computer mouse and a customized response button (accuracy 2 ms, response latency 9 ms) were used for response registration.

Prior to the experiment, participants filled in demographic and personality questionnaires: Neuroticism and Conscientiousness Scales of NEO PI-R (*Costa and McCrae, 1992*) and the BIS/BAS scale (*Strobel et al., 2001*), as well as a subset of the *Raven, 2000* progressive matrices as an index for figural-spatial intelligence. These measures were registered as potential control variables and for other purposes not addressed here. Participants were then seated in a shielded EEG cabin, where the experiment including EEG recording was conducted. Prior to the experiment proper, participants performed three practice trials.

## Psychophysiological recording and processing

Using BrainVision recorder software (Brain Products, München, Germany) we recorded EEG data from 64 Ag/AgCl electrodes mounted in an electrode cap (ECI Inc), referenced against Cz at a

sampling rate of 500 Hz. Electrodes below the eyes (IO1, IO2) and at the outer canthi (LO1, LO2) recorded vertical and horizontal ocular activity. We kept electrode impedance below 5 kΩ and applied a 100 Hz low pass filter, a time constant of 10 s, and a 50 Hz notch filter. At the beginning of the session we recorded 20 trials each of prototypical eye movements (up, down, left, right) for offline ocular artifact correction.

EEG data were processed using custom Matlab (The MathWorks Inc) scripts (*Frömer et al., 2018*) and EEGlab toolbox functions (*Delorme and Makeig, 2004*). We re-calculated to average reference and retrieved the Cz channel. The data were band pass filtered between 0.5 and 40 Hz. Ocular artifacts were corrected using BESA (*Ille et al., 2002*). We segmented the ongoing EEG from −200 to 800 ms relative to feedback onset. Segments containing artifacts were excluded from analyses, based on values exceeding ±150 μV and gradients larger than 50 μV between two adjacent sampling points. Baselines were corrected to the 200 ms pre-stimulus interval (feedback onset).

The FRN was quantified in single-trial ERP waveforms as peak-to-peak amplitude at electrode FCz, specifically as the difference between the minimum voltage in a window from 200 to 300 ms post-feedback onset and the preceding positive maximum in a window from −100 to 0 ms relative to the detected negative peak. To define the time windows for single-trial analyses of P3a and P3b amplitudes, we first determined the average subject-wise peak latencies at FCz and Pz, respectively, and exported 100 ms time windows centered on the respective latencies. Accordingly, the P3a was quantified on single trials as the average voltage within an interval from 330 to 430 ms after feedback onset across all electrodes within a fronto-central region of interest (ROI: F1, Fz, F2, FC1, FCz, FC2, C1, Cz, C2). P3b amplitude was quantified in single trials as the average voltage within a 416–516 ms interval post-feedback across all electrodes within a parietally-focused region of interest (ROI: CP1, CPz, CP2, P1, Pz, P2, PO3, POz, PO4).

## Analyses

Outlier inspection of the behavioral data identified one suspicious participant (average RT >10 s) and one trial each in four additional participants (RTs > 6 s, 0.4% of data of remaining participants). These data were excluded from further analyses. We computed two kinds of prediction errors (*Figure 3A*): SPE was determined as the absolute difference between predicted and actual interval length: |Prediction − Feedback|. RPE was computed as the difference between the absolute predicted error and the absolute actual error as revealed by feedback: |Prediction| − |Feedback|. We quantified confidence calibration as each participant's correlation of confidence and SPE (absolute deviation of the prediction from the actual outcome) across all trials, controlling for average error magnitude per block to account for shared changes in our confidence calibration measure with performance. To ease interpretation, we sign-reversed the correlations, such that higher values correspond to higher confidence calibration.

Statistical analyses were performed by means of linear mixed models (LMMs) using R (*R Development Core Team, 2014*) and the lme4 package (*R Package, 2014*). We chose LMMs, similar to linear multiple regression models, as they allow for parametric analyses of single-trial measures. Further, LMMs are robust to unequally distributed numbers of observations across participants, and simultaneously estimate fixed effects and random variance between participants in both intercepts and slopes. For all dependent variables, full models, including all predictors, were reduced step-wise until model comparisons indicated significantly decreased fit.

We report model comparisons and fit indices: Akaike Information Criterion (AIC) and Bayesian Information Criterion (BIC), which decrease with improving model fit. Random effect structures were determined using singular value decomposition. Variables explaining zero variance were removed from the random effects structure (*Bates et al., 2015*; *Matuschek et al., 2017*).

Prior to the analyses, error magnitude, RPE and SPE were scaled from ms to seconds and confidence and block were also scaled to a range of ±1 for similar scaling of all predictors. Furthermore, block, error magnitude, confidence, and SPE were centered on their medians for accurate intercept computation. RPE was not centered, as zero represents a meaningful value on the scale (predicted and actual error magnitude are the same), and positive and negative values are qualitatively different (negative and positive values represent outcomes that are, respectively, worse or better than expected). Model formulas are reported in the respective tables. Fixed effects are visualized using the effects package (*Fox and Weisberg, 2019*).

## Data availability

The datasets generated and analyzed during the current study are available under https://github.com/froemero/Outcome-Predictions-and-Confidence-Regulate-Learning (copy archieved at swh:1:rev:e8bfacf8fdb8126aade59581b98616b4f2fae7b3; *Frömer, 2021*).

## Code availability

Scripts for all analyses are available under https://github.com/froemero/Outcome-Predictions-and-Confidence-Regulate-Learning.

## Acknowledgements

We thank Lena Fliedner and Lara Montau for support in data acquisition and helpful discussions during the setup of the task, Rainer Kniesche for advice on programming the stimulus government, Markus Ullsperger, Adrian Haith, Martin Maier, and Rasha Abdel Rahman for valuable discussions, and Mehrdad Jazayeri for valuable feedback on a previous draft. RF is further grateful for the continuous scientific and personal support by her office mates at Humboldt-University, Benthe Kornrumpf and Florian Niefind, who made her life and work a lot more fun and happened to also have inspired the original title of this paper.

## Additional information

### Funding

| Funder | Grant reference number | Author |
| --- | --- | --- |
| NIH Office of the Director | R00 AG054732 | Matthew R Nassar |

The funders had no role in study design, data collection and interpretation, or the decision to submit the work for publication.

### Author contributions

Romy Frömer, Conceptualization, Data curation, Formal analysis, Investigation, Visualization, Methodology, Writing - original draft, Project administration, Writing - review and editing; Matthew R Nassar, Formal analysis, Supervision, Methodology, Writing - review and editing; Rasmus Bruckner, Formal analysis, Writing - review and editing; Birgit Stürmer, Writing - review and editing; Werner Sommer, Resources, Methodology, Writing - review and editing; Nick Yeung, Conceptualization, Supervision, Writing - review and editing

### Author ORCIDs

Romy Frömer https://orcid.org/0000-0002-9468-4014
Matthew R Nassar http://orcid.org/0000-0002-5397-535X
Rasmus Bruckner http://orcid.org/0000-0002-3033-6299

### Ethics

Human subjects: The study was performed following the guidelines of the ethics committee of the department of Psychology at Humboldt University. Participants gave informed consent to the experiment and were remunerated with course credits or 8€ per hour.

### Decision letter and Author response

Decision letter https://doi.org/10.7554/eLife.62825.sa1
Author response https://doi.org/10.7554/eLife.62825.sa2

## Additional files

### Supplementary files

- Supplementary file 1. Follow-up on prediction and performance precision effects on confidence.
- Supplementary file 2. Control analysis for confidence calibration effect on learning.
- Supplementary file 3. Follow-up on block and confidence effects on relative error signals.
- Supplementary file 4. Follow-up on confidence by block interaction on RPE benefit over error magnitude.
- Supplementary file 5. Block and confidence effects on error signals.
- Supplementary file 6. Follow-up on block and confidence effects on error signals.
- Supplementary file 7. Block and confidence effects on error signals.
- Supplementary file 8. Follow-up analyses on confidence-weighted predicted error magnitude effects on P3b.
- Supplementary file 9. Trial-to-trial improvements by block and previous error and modulations by previous P3b.
- Supplementary file 10. Follow-up on trial-to-trial improvements by block and previous error and modulations by previous P3b.
- Transparent reporting form

### Data availability

Scripts and source data for all analyses are available under https://github.com/froemero/Outcome-Predictions-and-Confidence-Regulate-Learning (copy archived at https://archive.softwareheritage.org/swh:1:rev:e8bfacf8fdb8126aade59581b98616b4f2fae7b3).

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
