## [Decision Letter]

**Acceptance summary:**

The authors show a novel and important finding that participants use self-knowledge to optimize learning. Participants in a time estimation task used post-response information about their temporal errors to optimize learning. This is evident in the neural prediction error signals that indexed deviations from the intended target response. This work nicely integrates reinforcement-learning, time estimation and performance monitoring.

**Decision letter after peer review:**

Thank you for submitting your article "I knew that! Response-based Outcome Predictions and Confidence Regulate Feedback Processing and Learning" for consideration by *eLife*. Your article has been reviewed by 3 peer reviewers, including Tadeusz Wladyslaw Kononowicz as the Reviewing Editor and Reviewer #1, and the evaluation has been overseen by Richard Ivry as the Senior Editor. The following individual involved in review of your submission has agreed to reveal their identity: Simon van Gaal (Reviewer #3).

The reviewers have discussed the reviews with one another and the Reviewing Editor has drafted this decision to help you prepare a revised submission.

Summary:

The authors tested 40 human volunteers in a time production task and post-production performance evaluation (with an initially unknown target duration and feedback scale) while recording EEG. The authors tested the hypothesis that confidence (both its absolute value and its calibration to performance) have an effect on learning and that it affects the processing of reward and sensory prediction errors.

The reviewers all found the results to be interesting and the work was well-conducted. At the same time the reviewers agreed that the authors should be able to address several issues and clarify multiple aspects of the task, performed analyses, and data interpretation. The comments were compiled into essential revisions where we summarize the remarks that should involve additional data analysis and those proposing changes in the manuscript.

Essential revisions:

Additional analyses:

1. The authors analyze correlations between Error Magnitude, Predicted Outcome and Confidence, however before proceeding to analysis of ERPs the manuscript could be improved by including similar analysis of confidence correlations with RPE and SPE, beyond the one relying only on Predicted Outcome (Table 1).

2. Related to point 1, panel 3A should belong to Figure 1, especially if analyses proposed in the first point are included.

3. The authors showed that Error Magnitude decreases on average. However, all ERP analyses were focused on the current trial. If these ERP signals indeed reflect some "updating of internal representations" they should have a relationship with the behavior or neural measures observed on the next trial. It would've been very interesting to see how the processing of feedback (in behavior and ERP responses) relates to performance on the next trial. These analyses should better support the claims of "updating of internal representations", which would considerably improve in impact and quality if these analyses will be reported.

4. Plausible changes of precision (variance) of temporal performance over the course of experiment. Variance dynamics across experimental session could affected the outcome of the confidence calibration. The authors rightfully show that Confidence Calibration was not related to Average Error Magnitude. The same check should be performed for Time Production variance. Moreover, the effects within participants and over the course of experiment should be considered and presumably included as covariates in the LMM.

5. Specific point from one of the reviewers: The authors mention again on page 24: "We also found that confidence modulated RPE effects on P3b amplitude, such that in initial blocks, where most learning took place, RPE effects were amplified for higher confidence, whereas this effect reversed in later blocks, where RPE effects were present for low, but not high confidence. This shift is intriguing and may indicate a functional change in feedback use as certainty in the response-outcome mapping increases and less about this mapping is learned from feedback, but the effect was not directly predicted and therefore warrants further research and replication." This is the one result where confidence interacts with other behavioral measures, in this case RPE, which is interesting, however it does so in an unpredicted and counterintuitive way. I wonder whether the authors can in some way get a better understanding of what's going on here? Possibly the paper by Colizoli et al. (2018, Sci Rep.) may be relevant. The authors here show how task difficulty (related to confidence) and error processing are reflecting in pupil size responses.

Other reviewer raised concerns on how different Confidence splits were computed. Although, the authors provide and an intriguing interpretation reference in the paragraph above, is it possible that the early and late effects originate in fact from different group of subjects?

To sum up, extending the analyses with respect to the interaction of confidence and RPE in modulation of P3b component would strongly benefit the manuscript.

6. There is not explicit statement on what exact instructions were given to participants beyond the following one: "participants were required to predict the feedback they would receive on each trial". The caption of Figure 1B says that "scaled relative to the feedback anchors". Therefore, it is not clear what was the primary objective of the task – accurate time production or predicting the feedback accurately? Participants could have increased time production variance to perform better on feedback prediction. If participants employed that kind of strategy that could have impact indices of learning from feedback.

Given the lack of clarity of what instruction was provided to participants it is still unclear on which aspect of the task the participants focused on in their learning. Error Magnitude decreases over trial, however does RPE and SPE increase over trials as well?

Reshaping the manuscript:

1. It was evident from all reviews that at many places an explicit link between interpretative statements and performed analyses were far from clear. Below we list a few specific examples:

– "Taken together, our findings provide evidence that feedback evaluation is an active constructive process that is fundamentally affected by an individual's internal representations of their own performance at the time of feedback." I wonder what results the authors refer to here and on what results this statement is based on.

– The authors say "In line with the notion that positive feedback is more informative than error feedback in motor learning, we, like others in the time estimation task (65,66), observed increasing P3b amplitude after more positive feedback, in addition to prediction error effect". It is not clear which outcome the authors are referring to. Is "better than expected" referred to as "positive feedback"? In this case "worse than expected" triggered higher P3b amplitude.

– On page 24 the authors conclude that "Learning was further supported by the adaptive regulation of uncertainty-driven updating via confidence." Although this sounds interesting I do not see the results supporting this conclusion (but maybe I have missed those). I also think this conclusion is rather difficult to follow. The sentence thereafter they say "Specifically, as deviations from the goal were predicted with higher confidence, these more precise outcome predictions enhanced the surprise elicited by a given prediction error. Thus, a notable finding revealed by our simulations and empirical data is that, counterintuitively, agents and participants learned more from feedback when confidence in their predictions had been high." Also here I have difficulty extracting what the authors really mean. What does it mean "surprise elicited by a prediction error"? To me these are two different measures, one signed one unsigned. Further, where is it shown that participants learn more from feedback when confidence in their prediction was high?

– Differences between blocks in the effect of confidence. This result is discussed twice: in the Results (p. 19) and Discussion. Only in the latter do the authors acknowledge that their interpretation of the effect is rather speculative. I would also flag that in Results, as it was neither part of the model predictions or their design.

2. Performed transformations involving confidence should be clearly explained.

3. Model specification (the formula) should be included in the table legend to aid readability and interpretation as it makes it immediately clear what was defined as a random or fixed effect.

4. On more conceptual level, the authors rely on the assumption that 'Feedback Prediction' is derived from efference copy, which carries motor noise only. In light of the goal of the current manuscript, that is an appropriate strategy. However, I think it should be acknowledged that in the employed paradigm part of behavioral variance may originate from inherent uncertainty of temporal representations (Balci, 2011). Typically, time production variance is partition into a 'clock' variance and 'motor' variance. I have a feeling that this distinction should be spelled out in the manuscript and if assumptions are made they shall be spelled out clearer. Moreover, recent work attempted to tease apart origins of 'Feedback Predictions', indicating that it is unlikely that they originate solely from motor variability (Kononowicz and Van Wassenhove, 2019).

5. The main predictions of the experiment are described in the first paragraph of the Results. But they are not reflected in Figure 1, which is referenced in that paragraph. I would have expected an illustration of the effects of confidence, and instead that only appears on Figure 2. The authors have clear predictions that drive the analysis, but this is not reflected in the flow of the text.

6. Simulations (Figure 2. B, D): As far as I can tell, the model does not capture the data in two ways: it fails to address the cross-over effect (which the authors address) but also does not account for the apparent tendency of the data to increase the error on later trials (whereas the model predict a strict decrease in error over the course of the experiment). The second aspect is not addressed in the Discussion, I think (or I missed it). Do the authors think this is just fatigue, and therefore not consider it as a reason to modify the model? Also Panels 2.A. And C do not really match in the sense that the simulation is done over a much wider range of predicted outcomes. It seems like the model parameters were not fine-tuned to the data. Perhaps this is not strictly necessary if the quantitative predictions of the effects of confidence remain unchanged with a narrower range, but it is perhaps worth discussing.

7. "… it is unknown whether reward prediction errors signaled by the FRN rely on predictions based only on previous feedback, or whether they might incorporate ongoing performance monitoring". I think that phrase should be rephrased based on the findings of Miltner et al. (1997), correctly cited in the manuscript, which showed that FRN was responsive to correct and incorrect feedback in time estimation.

8. Relevance of the dart-throwing example: In the task, participants initially had no idea about the length of the to-be-reproduced interval, and instead had to approximate it iteratively. It was not immediately clear to me how this relates to a dart-throw, where the exact target position is known. I think I understand the authors that the unknown target here is "internal" – the specific motor commands that would lead to a bulls-eye are unknown and only iteratively approximated. If that interpretation is correct, I would recommend the authors clarify it explicitly in the paper, to aid the reader to make a connection. Or perhaps I misunderstood it. Either way it would be important to clarify it.

Balci, F., Freestone, D., Simen, P., Desouza, L., Cohen, J. D., and Holmes, P. (2011). Optimal temporal risk assessment. Front Integr Neurosci, 5, 56.

Colizoli, O., De Gee, J. W., Urai, A. E., and Donner, T. H. (2018). Task-evoked pupil responses reflect internal belief states. Scientific reports, 8(1), 1-13.

Correa, C. M., Noorman, S., Jiang, J., Palminteri, S., Cohen, M. X., Lebreton, M., and van Gaal, S. (2018). How the level of reward awareness changes the computational and electrophysiological signatures of reinforcement learning. Journal of Neuroscience, 38(48), 10338-10348.

Kononowicz, T. W., and Van Wassenhove, V. (2019). Evaluation of Self-generated Behavior: Untangling Metacognitive Readout and Error Detection. Journal of cognitive neuroscience, 31(11), 1641-1657.

---

## [Author Response]

Essential revisions:Additional analyses:1. The authors analyze correlations between Error Magnitude, Predicted Outcome and Confidence, however before proceeding to analysis of ERPs the manuscript could be improved by including similar analysis of confidence correlations with RPE and SPE, beyond the one relying only on Predicted Outcome (Table 1).

Thank you for this recommendation. We agree that our claim that Confidence reflects the precision of predictions can be tested more directly. As a critical test of our assumption that confidence varies with the precision of the prediction – i.e. SPE, we now analyze Confidence as the dependent variable and test how it relates to the precision of the prediction (sensory prediction error), the precision of performance (error magnitude), and how these relationships change across blocks. Consistent with our theoretical assumption, we find a robust relationship with SPE. We also find that Confidence increases with increasing error magnitude, and more so in later blocks. The latter finding is important because it shows that participants were in fact reporting their confidence in the accuracy of their predictions and not confidence in their performance.

The novel Results section reads as follows: “To test more directly our assumption that Confidence tracks the precision of predictions, we followed up on these findings with a complementary analysis of Confidence as the dependent variable and tested how it relates to the precision of predictions (absolute discrepancy between predicted and actual outcome, see sensory prediction error, SPE below), the precision of performance (error magnitude), and how those change across blocks (Table 2). […] This pattern is thus not consistent with learning. Importantly, whereas error magnitude was robustly related to confidence only in the last two blocks, the precision of predictions was robustly related to confidence throughout.”

As to the relationship with RPE, we agree that this is an important relationship to look at, particularly given the somewhat surprising 3-way interaction on the P3b (point 5). We think that in the context of feedback processing and ERPs this relationship is the most relevant and informative and therefore we now introduce a novel set of analyses that specifically investigates changes in our feedback regressors (RPE, SPE and EM) over time and their interaction with confidence.

The novel section reads as follows: “At the core of our hypothesis and model lies the change in feedback processing as a function of outcome predictions and confidence. […] Accordingly, we predicted that participants’ internal evaluations would modulate feedback processing as indexed by distinct feedback-related potentials in the EEG: the feedback-related negativity (FRN), P3a and P3b.”

2. Related to point 1, panel 3A should belong to Figure 1, especially if analyses proposed in the first point are included.

We agree that foreshadowing the different dimensions along which feedback can be evaluated, would help the readers. We have now altered Figure 1 to include the dissociation between performance and prediction and how one being better (or worse) than the other can alter the subjective valence of the outcome. In order to maintain continuity of Figure 1 we have introduced these concepts as part of the cartoon example, rather than in terms of our exact task. Thus, we still include panel 3A in a separate figure in the new subsection we added following your recommendation where we unpack the different kinds of prediction errors in our task, how they change across blocks and as a function of confidence. We hope that this provides the relevant information at the appropriate locations in the manuscript.

3. The authors showed that Error Magnitude decreases on average. However, all ERP analyses were focused on the current trial. If these ERP signals indeed reflect some "updating of internal representations" they should have a relationship with the behavior or neural measures observed on the next trial. It would've been very interesting to see how the processing of feedback (in behavior and ERP responses) relates to performance on the next trial. These analyses should better support the claims of "updating of internal representations", which would considerably improve in impact and quality if these analyses will be reported.

We agree that it would be great if we could link the ERPs to adjustments in behavior and we have now added an exploratory analysis to link P3b to trial-by-trial adjustments to feedback. To show such trial-by-trial adjustments, we quantify the degree to which the performance improvement on the current trial relates to the error on the previous trial and demonstrate that this relationship is contingent on P3b amplitude, specifically in the first block when most learning takes place. We set up a model for improvement on the current trial (previous error minus current error) as a function of error magnitude on the previous trial in interaction with P3b amplitude and their interaction with block. We found the expected 3-way interaction (see Figure S4). When participants’ performance improves the most (in block 1), larger P3b amplitudes to the feedback on the previous trial lead to larger improvements on the current trial. Note, however, that this finding is mostly driven by large errors and that participants are overall likely to perform worse following smaller errors. Responses at the single trial level are subject to substantial noise – the very prerequisite for our study – likely masking local adjustments in the underlying representations. Thus, we are wary to overinterpret this result and thus highlight potential caveats to this analysis in a new section we now added. –

The new section reads: “We next explored whether P3b amplitude is associated with trial-by-trial adjustments. […] While intriguing and in line with previous work linking P3b to trial-by-trial adjustments in behavior, these results should be interpreted with a degree of caution given that the present task is not optimized to test for trial-totrial adjustments in behavior.”

4. Plausible changes of precision (variance) of temporal performance over the course of experiment. Variance dynamics across experimental session could affected the outcome of the confidence calibration. The authors rightfully show that Confidence Calibration was not related to Average Error Magnitude. The same check should be performed for Time Production variance. Moreover, the effects within participants and over the course of experiment should be considered and presumably included as covariates in the LMM.

This is an excellent point. We addressed this concern in two ways as suggested by the reviewer: First, we computed the correlation between participants’ response variance and their confidence calibration. Second, to capture changes over time, we added participants running average response variance as a covariate to the model of error magnitude.

In the manuscript we now write: “Confidence calibration was also not correlated with individual differences in response variance (r = – 2.07e-4, 95%CI = [-0.31, 0.31], p =.999), and the interaction of confidence calibration and block was robust to controlling for running average response variance (Supplementary File 2).”

5. Specific point from one of the reviewers: The authors mention again on page 24: "We also found that confidence modulated RPE effects on P3b amplitude, such that in initial blocks, where most learning took place, RPE effects were amplified for higher confidence, whereas this effect reversed in later blocks, where RPE effects were present for low, but not high confidence. This shift is intriguing and may indicate a functional change in feedback use as certainty in the response-outcome mapping increases and less about this mapping is learned from feedback, but the effect was not directly predicted and therefore warrants further research and replication." This is the one result where confidence interacts with other behavioral measures, in this case RPE, which is interesting, however it does so in an unpredicted and counterintuitive way. I wonder whether the authors can in some way get a better understanding of what's going on here?

We agree that this three-way interaction deserves more unpacking, particularly given the relevance of interactions with confidence for our theoretical hypothesis. The new analyses in response to comments 1 and 2 made it clear that changes in RPE effects with Confidence and Block are complicated by the fact that RPE is not systematically related to Confidence, nor the degree to which it reduces error signals relative to error magnitude. The interaction could reflect changes in the components of RPE – Error Magnitude and Predicted Error Magnitude.

To address this point, we now report a complementary analysis that uses predicted error magnitude rather than RPE. This has the advantage that it allows us to test the specific prediction that predictions are weighted by confidence when processing feedback. This is exactly what we find. In particular in the first block, when most learning takes place, the degree to which predicted errors are discounted (as reflected in a decrease in P3b amplitude) depends on Confidence, and higher confidence is overall associated with larger P3b amplitudes. In later blocks, main effects of predicted error magnitude emerge (and we know from prior analyses, that performance is more variable when confidence is low in those late blocks allowing for larger errors to discount), likely underlying the late confidence by RPE interaction in our original analysis.

The novel Results section reads as follows: “Our hypothesis states that the degree to which people rely on their predictions when learning from feedback should vary with their confidence in those predictions. […] In later trials however, when confidence is higher overall, participants discount their predicted errors even when confidence is relatively low.”

The corresponding section in the discussion now reads: “We also found that confidence modulated the degree to which predicted error magnitude reduced P3b amplitude, such that in initial blocks, where most learning took place, predicted error magnitude effects were amplified for higher confidence, whereas this effect diminished in later blocks, where predicted error magnitude effects were present also for low confidence (and performance and prediction errors were attenuated when confidence was high).”

Possibly the paper by Colizoli et al. (2018, Sci Rep.) may be relevant. The authors here show how task difficulty (related to confidence) and error processing are reflecting in pupil size responses.

Thank you for pointing out the Colizoli reference to us. This is indeed very relevant and we now cite it in the discussion.

Other reviewer raised concerns on how different Confidence splits were computed. Although, the authors provide and an intriguing interpretation reference in the paragraph above, is it possible that the early and late effects originate in fact from different group of subjects?

The confidence splits in the original analysis were merely performed to get a sense of the underlying pattern. We have now removed these follow-up analyses, as we follow up on the 3-way interaction as described above. The pattern in these novel analyses renders a between-group effect unlikely. When separating out Confidence mean and z-scored variations for each participant, we find that the within-subject variability drives the effects we observe – reassuring us about our interpretation. However, confidence levels between subjects seem important as well, as results become unstable when they are not included, maybe because confidence changes with learning.

To sum up, extending the analyses with respect to the interaction of confidence and RPE in modulation of P3b component would strongly benefit the manuscript.

We agree that the extension of the analyses has benefitted the manuscript and thank the reviewers for their recommendation.

6. There is not explicit statement on what exact instructions were given to participants beyond the following one: "participants were required to predict the feedback they would receive on each trial". The caption of Figure 1B says that "scaled relative to the feedback anchors". Therefore, it is not clear what was the primary objective of the task – accurate time production or predicting the feedback accurately? Participants could have increased time production variance to perform better on feedback prediction. If participants employed that kind of strategy that could have impact indices of learning from feedback.Given the lack of clarity of what instruction was provided to participants it is still unclear on which aspect of the task the participants focused on in their learning. Error Magnitude decreases over trial, however does RPE and SPE increase over trials as well?

Participants were instructed to learn to produce the correct the time interval. Thus, the emphasis was on correct timing production. In addition, they were asked to estimate the error in their response.

We now clarify in the Method: “Participants were instructed that their primary goal in this task is to learn to produce an initially unknown time interval. In addition, they were asked to predict the direction and magnitude of any errors they produced and their confidence in those predictions.”

As now reported in the manuscript, SPE decreased over the course of the experiment just like Error magnitude (primarily from block 1 to 2). Changes in RPE are difficult to interpret given that both better than expected and worse than expected outcomes are still “incorrectly” predicted. SPE is thus clearly the superior indicator. However, we can also look at changes in absolute RPE across blocks and we find that, like Error Magnitude and SPE, it decreases across blocks and primarily from block 1 to 2. Note, however, that these changes are primarily driven by improvements in time estimation performance and diminish substantially once we control for Error Magnitude. We have now added all these analyses in the Feedback section prior to the ERP analyses.

Reshaping the manuscript:1. It was evident from all reviews that at many places an explicit link between interpretative statements and performed analyses were far from clear. Below we list a few specific examples:– "Taken together, our findings provide evidence that feedback evaluation is an active constructive process that is fundamentally affected by an individual's internal representations of their own performance at the time of feedback." I wonder what results the authors refer to here and on what results this statement is based on.

We can see how some of this phrasing goes beyond the key findings of our study. We have now simplified the sentence to more distinctly reflect our contributions: “Taken together, our findings provide evidence that feedback evaluation is fundamentally affected by an individual’s internal representations of their own performance at the time of feedback.”

– The authors say "In line with the notion that positive feedback is more informative than error feedback in motor learning, we, like others in the time estimation task (65,66), observed increasing P3b amplitude after more positive feedback, in addition to prediction error effect". It is not clear which outcome the authors are referring to. Is "better than expected" referred to as "positive feedback"? In this case "worse than expected" triggered higher P3b amplitude.

Thank you, we now realize that this was ambiguous. This statement refers to objective performance and we have now changed the statement to make this clear. “In line with the notion that in motor learning feedback about success is more informative than feedback about failure, we, like others in the time estimation task ^66,67^, observed increasing P3b amplitude after feedback about more accurate performance (i.e. for smaller error magnitude), in addition to prediction error effects.”

– On page 24 the authors conclude that "Learning was further supported by the adaptive regulation of uncertainty-driven updating via confidence." Although this sounds interesting I do not see the results supporting this conclusion (but maybe I have missed those). I also think this conclusion is rather difficult to follow. The sentence thereafter they say "Specifically, as deviations from the goal were predicted with higher confidence, these more precise outcome predictions enhanced the surprise elicited by a given prediction error. Thus, a notable finding revealed by our simulations and empirical data is that, counterintuitively, agents and participants learned more from feedback when confidence in their predictions had been high." Also here I have difficulty extracting what the authors really mean. What does it mean "surprise elicited by a prediction error"? To me these are two different measures, one signed one unsigned. Further, where is it shown that participants learn more from feedback when confidence in their prediction was high?

It seems that there are some misunderstandings here that we have now tried to clarify. To address the unclear link between the conclusion and our findings, we have now extended this section to read: “Learning – in our computational model and in our participants – was further supported by the adaptive regulation of uncertainty-driven updating via confidence. […] Thus, a notable finding revealed by our simulations and empirical data is that, counterintuitively, agents and participants learned *more* from feedback when confidence in their predictions had been high.”

We believe it is important to dissociate between prediction errors and surprise. In particular, we quantify two types of prediction errors, where only RPE is signed (better or worse than predicted) and SPE (how different than predicted) is not. However, we propose that surprise is even more nuanced than the latter, because it is not only dependent on the absolute mismatch between prediction and outcome, but also on the confidence with which the prediction was made. That is what we simulate using Shannon information. To make this more apparent from the beginning and provide intuitions, we now foreshadow this concept in the introduction: “In the throwing example above, the more confident you are about the exact landing position of the dart, the more surprised you should be when you find that landing position to be different: The more confident you are, the more evidence you have that your internal model linking angles to landing positions is wrong, and the more information you get about how this model is wrong. […] However, this reasoning assumes that your predictions are in fact more precise when you are more confident, i.e., that your confidence is well calibrated (Figure 1B).”

We have further altered the relevant sentences in last paragraph in the introduction to read: “That is to say, an error that could be predicted based on internal knowledge of how an action was executed should *not* yield a large surprise (P3a) or reward prediction error (FRN) signal in response to an external indicator of the error (feedback). However, any prediction error should be *more* surprising when predictions were made with higher confidence.”

– Differences between blocks in the effect of confidence. This result is discussed twice: in the Results (p. 19) and Discussion. Only in the latter do the authors acknowledge that their interpretation of the effect is rather speculative. I would also flag that in Results, as it was neither part of the model predictions or their design.

Thank you for pointing out this oversight on our end. We have now entirely removed the interpretation of the 3-way interaction from the Results section. As you can see described in our response to point 5 above we have added extensive additional analyses that provide better insights into the confidence effects across blocks as they relate to our hypothesis and now rely on these additional findings for our interpretations instead.

2. Performed transformations involving confidence should be clearly explained.

We assume that this comment refers to the computation of confidence calibration. The reviewers are right in that we did not clearly explain that in the results. Now that we added the novel group-level analysis of the underlying relationship, we build on that to unpack more clearly how we derive the individual difference

measure before moving on to the section where we test for differences in learning varying with confidence calibration.

“Having demonstrated that, across individuals, confidence reflects the precision of their predictions (via the correlation with SPE), we next quantified this relationship for each participant separately as an index of their confidence calibration. […] We next tested our hypothesis that confidence calibration relates to learning.”

3. Model specification (the formula) should be included in the table legend to aid readability and interpretation as it makes it immediately clear what was defined as a random or fixed effect.

We have now added formulas to all tables.

4. On more conceptual level, the authors rely on the assumption that 'Feedback Prediction' is derived from efference copy, which carries motor noise only. In light of the goal of the current manuscript, that is an appropriate strategy. However, I think it should be acknowledged that in the employed paradigm part of behavioral variance may originate from inherent uncertainty of temporal representations (Balci, 2011). Typically, time production variance is partition into a 'clock' variance and 'motor' variance. I have a feeling that this distinction should be spelled out in the manuscript and if assumptions are made they shall be spelled out clearer. Moreover, recent work attempted to tease apart origins of 'Feedback Predictions', indicating that it is unlikely that they originate solely from motor variability (Kononowicz and Van Wassenhove, 2019).

First of all, apologies for having missed these papers. Thank you for pointing them out to us! It’s true, we previously used motor noise as a blanket term to account for multiple sources of variability. This was more of a convenience than a strong assumption. We have now replaced motor noise with response noise throughout the manuscript and briefly mention the two different drivers, citing the mentioned papers. “First, errors signaled by feedback include contributions of response noise, e.g. through variability in the motor system or in the representations of time ^24,41^. Second, the efference copy of the executed response (or the estimate of what was done) varies in its precision.”

5. The main predictions of the experiment are described in the first paragraph of the Results. But they are not reflected in Figure 1, which is referenced in that paragraph. I would have expected an illustration of the effects of confidence, and instead that only appears on Figure 2. The authors have clear predictions that drive the analysis, but this is not reflected in the flow of the text.

Thank you for your comment. We agree that it would help to visualize our predictions in the beginning. We have now revised Figure 1 to clarify key concepts and show our main predictions. We still show the model predictions and the empirical data as tests of these predictions in Figure 2.

6. Simulations (Figure 2. B, D): As far as I can tell, the model does not capture the data in two ways: it fails to address the cross-over effect (which the authors address) but also does not account for the apparent tendency of the data to increase the error on later trials (whereas the model predict a strict decrease in error over the course of the experiment). The second aspect is not addressed in the Discussion, I think (or I missed it). Do the authors think this is just fatigue, and therefore not consider it as a reason to modify the model? Also Panels 2.A. And C do not really match in the sense that the simulation is done over a much wider range of predicted outcomes. It seems like the model parameters were not fine-tuned to the data. Perhaps this is not strictly necessary if the quantitative predictions of the effects of confidence remain unchanged with a narrower range, but it is perhaps worth discussing.

To follow up on this, we plotted the log transformed running average error magnitude for three bins of confidence calibration. As can be seen in Figure 2—figure supplement 3, our statistical model approximates, but does not properly capture the shape of the learning curves, which rather seems to saturate within the first 100 trials for low confidence calibration, than showing a marked increase towards the end. We now note this in the figure caption. This figure shows the running average log-transformed error magnitude (10 trials) averaged within Confidence calibration terciles across trials. Computing running averages was necessary to denoise the raw data for display. The edited figure caption reads: “Note that the combination of linear and quadratic effects approximates the shape of the learning curves, better than a linear effect alone, but predicts an exaggerated uptick in errors towards the end, cf. Figure 2 – supplement 3.”

7. "… it is unknown whether reward prediction errors signaled by the FRN rely on predictions based only on previous feedback, or whether they might incorporate ongoing performance monitoring". I think that phrase should be rephrased based on the findings of Miltner et al. (1997), correctly cited in the manuscript, which showed that FRN was responsive to correct and incorrect feedback in time estimation.

We think this is a misunderstanding. As the Reviewer describes, Miltner et al. demonstrated that on average, error feedback elicits a negative deflection over fronto-central sites (FRN) relative to correct feedback. They do not consider expectations/predictions at all – either based on performance history or on performance monitoring. This finding was later built on and extended (Holroyd and Coles, 2002) by showing that the processing of error and correct feedback is sensitive to contextual expectations, i.e. reflects reward prediction error, not just error feedback per se. We extend this line of work further, by asking whether beyond the contextually-defined reward prediction error, FRN amplitude is sensitive to response-based outcome predictions derived through internal monitoring. Thus, the key question in our paper is whether error detection feeds into the prediction that underlies the prediction error processing reflected in the FRN. Miltner et al. have not shown or tested that. To avoid confusion, we have now changed the corresponding sentence to read: “However, it is unknown whether reward prediction errors signaled by the FRN contrast current feedback with predictions based only on previous (external) feedback, or whether they might incorporate ongoing (internal) performance monitoring.”

8. Relevance of the dart-throwing example: In the task, participants initially had no idea about the length of the to-be-reproduced interval, and instead had to approximate it iteratively. It was not immediately clear to me how this relates to a dart-throw, where the exact target position is known. I think I understand the authors that the unknown target here is "internal" – the specific motor commands that would lead to a bulls-eye are unknown and only iteratively approximated. If that interpretation is correct, I would recommend the authors clarify it explicitly in the paper, to aid the reader to make a connection. Or perhaps I misunderstood it. Either way it would be important to clarify it.

Thank you, yes, that is exactly right. One can think of the feedback scale just like the target. In each case, the right movement that produces the desired outcome needs to be learned. It doesn’t matter if I know that the target interval is 1.5 s if I don’t have a good sense of what that means for my time production. Similarly, it doesn’t help me to know where the target is if I don’t know how to reach it. Thus, the reviewer is exactly right that what is being iteratively approximated is the correct response. We now unpack the dart-throwing example in more detail throughout the introduction and when we introduce the task. To explicitly tie the relevant concepts together we now write: “In comparison to darts throwing as used in our example, the time estimation task requires a simple response – a button press – such that errors map onto a single axis that defines whether the response was provided too early, timely, or too late and by how much. These errors can be mapped onto a feedback scale and, just as in the darts example where one learns the correct angle and acceleration to hit the bullseye, participants here can learn the target timing interval.”

Balci, F., Freestone, D., Simen, P., Desouza, L., Cohen, J. D., and Holmes, P. (2011). Optimal temporal risk assessment. Front Integr Neurosci, 5, 56.Colizoli, O., De Gee, J. W., Urai, A. E., and Donner, T. H. (2018). Task-evoked pupil responses reflect internal belief states. Scientific reports, 8(1), 1-13.Correa, C. M., Noorman, S., Jiang, J., Palminteri, S., Cohen, M. X., Lebreton, M., and van Gaal, S. (2018). How the level of reward awareness changes the computational and electrophysiological signatures of reinforcement learning. Journal of Neuroscience, 38(48), 10338-10348.Kononowicz, T. W., and Van Wassenhove, V. (2019). Evaluation of Self-generated Behavior: Untangling Metacognitive Readout and Error Detection. Journal of cognitive neuroscience, 31(11), 1641-1657.